# Dynamic interactions between E-cadherin and Ankyrin-G mediate epithelial cell polarity maintenance

Chao Kong[1,2], Xiaozhan Qu[2,3], Mingming Liu[1], Weiya Xu[2,3], Da Chen[1,2], Yanshen Zhang[1], Shan Zhang[1], Feng Zhu[3], Zhenbang Liu[3], Jianchao Li[4], Chengdong Huang [2,3] ✉ & Chao Wang [1,2] ✉

E-cadherin is an essential cell–cell adhesion protein that mediates canonical cadherin-catenin complex formation in epithelial lateral membranes. Ankyrin-G (AnkG), a scaffold protein linking membrane proteins to the spectrin-based cytoskeleton, coordinates with E-cadherin to maintain epithelial cell polarity. However, the molecular mechanisms governing this complex formation and its relationships with the cadherin-catenin complex remain elusive. Here, we report that AnkG employs a promiscuous manner to encapsulate three discrete sites of E-cadherin by the same region, a dynamic mechanism that is distinct from the canonical 1:1 molar ratio previously described for other AnkG or E-cadherin-mediated complexes. Moreover, we demonstrate that AnkG-binding-deficient E-cadherin exhibited defective accumulation at the lateral membranes and show that disruption of interactions resulted in cell polarity malfunction. Finally, we demonstrate that E-cadherin is capable of simultaneously anchoring to AnkG and β-catenin, providing mechanistic insights into the functional orchestration of the ankyrin-spectrin complex with the cadherin-catenin complex. Collectively, our results show that complex formation between E-cadherin and AnkG is dynamic, which enables the maintenance of epithelial cell polarity by ensuring faithful targeting of the adhesion molecule-scaffold protein complex, thus providing molecular mechanisms for essential E-cadherin-mediated complex assembly at cell–cell junctions.

Cells are connected to form tight, organized multicellular structures in multicellular organisms[1–4]. Adherens junctions, characterized by the cell adhesion molecule E-cadherin, are a defining feature of all epithelial cells; these junctions enhance the mechanical stability of cells, thus playing crucial roles in the stability and organization of epithelial lateral membranes and the maintenance of cell polarity[5,6]. The anchoring of E-cadherin to plasma membranes is essential for physiological functions and is precisely regulated by tethering to the underlying cytoskeleton via two principal pathways: in the first scenario, E-cadherin binds to β-catenin through its C-terminal domain,

[1]Department of Neurology, the First Affiliated Hospital of USTC, Ministry of Education Key Laboratory for Membrane-less Organelles & Cellular Dynamics, Center for Advanced Interdisciplinary Science and Biomedicine of IHM, Hefei National Research Center for Physical Sciences at the Microscale, School of Life Sciences, Division of Life Sciences and Medicine, University of Science and Technology of China, Hefei, China. [2]Biomedical Sciences and Health Laboratory of Anhui Province, University of Science and Technology of China, Hefei, China. [3]Ministry of Education Key Laboratory for Membrane-less Organelles & Cellular Dynamics, Hefei National Research Center for Physical Sciences at the Microscale, School of Life Sciences, Division of Life Sciences and Medicine, University of Science and Technology of China, Hefei, China. [4]Division of Cell, Developmental and Integrative Biology, School of Medicine, South China University of Technology, Guangzhou, China. ✉e-mail: huangcd@ustc.edu.cn; cwangust@ustc.edu.cn

which recruits α-catenin, thus forming the canonical cadherin-catenin complex to connect with the actin-based cytoskeleton[7–9]; in the second scenario, E-cadherin directly binds to Ankyrin-G (AnkG), a scaffold protein linking membrane proteins to the spectrin-based cytoskeleton, to be stabilized at the lateral membranes of epithelial cells[10,11]. Loss of E-cadherin at the epithelial cell membrane is a defining characteristic of the epithelial-mesenchymal transition[12] and promotes tumor cell metastasis via multiple downstream signaling pathways, including β-catenin-mediated Wnt signaling[13]. However, the relationships between these two defining pathways for E-cadherin membrane localization are predominantly unknown.

AnkG belongs to a family of scaffold protein ankyrins, consisting of three family members, AnkR, AnkB, and AnkG, which are known for their recognition of diverse membrane proteins through their N-terminal 24 ankyrin repeats (ANK repeats). Mechanistically, ankyrins can utilize a combination of multiple binding sites in the inner groove surface of their ANK repeats to interact with disordered peptides present in various membrane-localized target proteins, with a stoichiometry usually at a 1:1 ratio[14–16]. The elongated conformation of the ANK repeats from ankyrins endows itself with an ideal binding module for binding with intrinsically disordered peptides present on membrane proteins, including for example, the known binding partners Nav1.2, KCNQ2/3, and Neurofascin in the nervous system[17–19]. AnkG has also been reported to bind with the juxtamembrane region of E-cadherin and is associated with the spectrin-actin network through binding to β2-spectrin, thus promoting the steady-state polarity of E-cadherin by retaining these proteins on the lateral membranes[20,21]. In AnkG knockout mice, E-cadherin was apically mis-sorted, and the lateral membrane heights of the collecting duct epithelial cells and bronchial epithelial cells were significantly reduced, indicating that AnkG is necessary for E-cadherin's stability at the lateral membranes[22]. However, the detailed molecular basis of AnkG-E-cadherin complex formation is still poorly understood.

Previous studies have shown that a highly conserved motif in the E-cadherin juxtamembrane region can bind with the p120 catenin armadillo repeat domain and have reported both dynamic and static interactions mediated by the two domains[23]. The crystal structures of the β-catenin armadillo repeat domain in complex with the E-cadherin C-terminal domain (unphosphorylated and phosphorylated) have also been solved[24]. Interestingly, the E-cadherin-β-catenin interaction surface is relatively extensive, involving the C-terminal ~100 residues of the E-cadherin cytoplasmic domain and the entire β-catenin armadillo repeat domain, forming a definite 1:1 complex. Thus, a fundamental question about E-cadherin is how E-cadherin coordinates such diverse complexes (e.g., AnkG, β-catenin, p120 catenin) through its ~150 residue cytoplasmic tail (Fig. 1a). A general strategy E-cadherin may utilize is that it employs different binding motifs for distinct binders. However, whether these E-cadherin-mediated complexes are mutually exclusive or form functional complexes simultaneously has not yet been investigated. Therefore, dissection of the detailed molecular mechanisms governing E-cadherin complex formation and elucidation of its relationships could deepen our understanding of adherens junction formation and cell polarity maintenance.

In the present study, we found that, distinct from canonical 1:1 binding for either the AnkG ANK repeats-mediated complex or E-cadherin-catenin complex, three discrete segments of the E-cadherin C-terminal domain (CTD) can bind to the same region of the inner groove formed by AnkG ANK repeats. The full-length E-cadherin CTD binds to AnkG ANK repeats with a molar ratio in the range of 1:1 to 1:2, thus forming a dynamic complex of E-cadherin–AnkG. We further found that hydrophobic interactions involving aromatic residues of the three binding sites from E-cadherin are essential for the assembly of this complex. We demonstrate dynamic complex formation through methyl-TROSY NMR analysis. Using MDCK cell cultures, we show that AnkG-binding-deficient E-cadherin exhibited defective accumulation

at the lateral membranes and demonstrate that disruption of E-cadherin-AnkG complex formation results in lateral membrane organization defects. Finally, we show that the E-cadherin-AnkG complex retains the β-catenin interaction, which connects ankyrin-spectrin with the canonical cadherin-catenin complex. Collectively, these findings expand the known types of interactions mediated by both ANK repeats of ankyrins and E-cadherin, conceptually linking the detected dynamic interactions with cell polarity maintenance.

## Results

### The E-cadherin C-terminal domain binds to AnkG R1-24

E-cadherin is a membrane-spanning protein with three domains: the N-terminal extracellular tandem repeats responsible for cell–cell adhesion, a single transmembrane helix, and a cytoplasmic C-terminal domain (residues 734-884, hereafter named CTD) that interacts with diverse intracellular partners (Fig. 1a). Given that the E-cadherin CTD was reported to bind with AnkG[20], we first used purified E-cadherin CTD and AnkG 24 ANK repeat (residues 38-855, hereafter named R1-24) proteins to perform isothermal titration calorimetry (ITC) assays to evaluate the binding affinity between E-cadherin and AnkG. The dissociation constant ($K_d$) for the binding of the full-length E-cadherin CTD to AnkG R1-24 was ~1.3 μM (Fig. 1b). The apparent N value was approximately 0.22, suggesting that the binding stoichiometry of E-cadherin and AnkG might not be 1:1; this would be distinct from the 1:1 binding previously observed for AnkG R1-24 recognition of other targets (Fig. 1b). We also conducted analytical gel filtration chromatography (AGFC) assays that showed that the E-cadherin CTD can directly interact with AnkG R1-24 (Fig. 1c). We mixed E-cadherin CTD and AnkG R1-24 at a 1:1 molar ratio: they formed a complex peak, although it was conspicuous that a substantial portion of the E-cadherin CTD was detected as unbound (Fig. 1c). Beyond indicating that the 1:1 molar ratio does not saturate E-cadherin binding, these AGFC data offer a second line of experimental evidence (bolstering our ITC data) that AnkG R1-24's recognition of the E-cadherin CTD is apparently distinct from its previously reported interactions with other binding targets.

We next used an NMR-based approach to investigate E-cadherin CTD binding with AnkG R1-24. NMR spectra of the E-cadherin CTD feature a single and coherent set of resonance lines for the backbone amide moieties (Fig. 1d, blue), which enabled near-complete backbone NMR assignments. Secondary assessments by chemical shift analysis[25], together with the narrow NMR chemical shift dispersions (Fig. 1d, blue), indicated that the E-cadherin CTD in solution adopts a random coil-like structure (Fig. 1e), consistent with previous studies reporting that the E-cadherin CTD showed no obvious secondary structure[26].

We also performed NMR titrations to identify which region(s) within the E-cadherin CTD are recognized by AnkG. Multiple-step titrations of unlabeled AnkG R1-24 into the isotopically labeled E-cadherin CTD led to decreases in the amide peak intensities with prominent dips for two regions: residues 751-777 and 829-867 (Fig. 1d, f). Upon AnkG titration, only a few residues exhibit noticeable chemical shift perturbations (CSPs) (Fig. 1d, red and yellow) with some extent of line-broadening (Fig. 1d). Collectively, these results allow us to identify two regions of the E-cadherin CTD that mediate the AnkG interaction (Fig. 1f). Moreover, our NMR data indicate that the E-cadherin CTD remains unstructured upon binding with AnkG and that the two proteins apparently form a dynamic complex that does not settle in a static conformation.

### Discrete sites within E-cadherin bind to AnkG in a dynamic manner

To map out the precise binding sites between E-cadherin and AnkG, we next conducted ITC analyses using various truncation fragments from the E-cadherin CTD titrated to AnkG R1-24 (Fig. S1). We initially divided

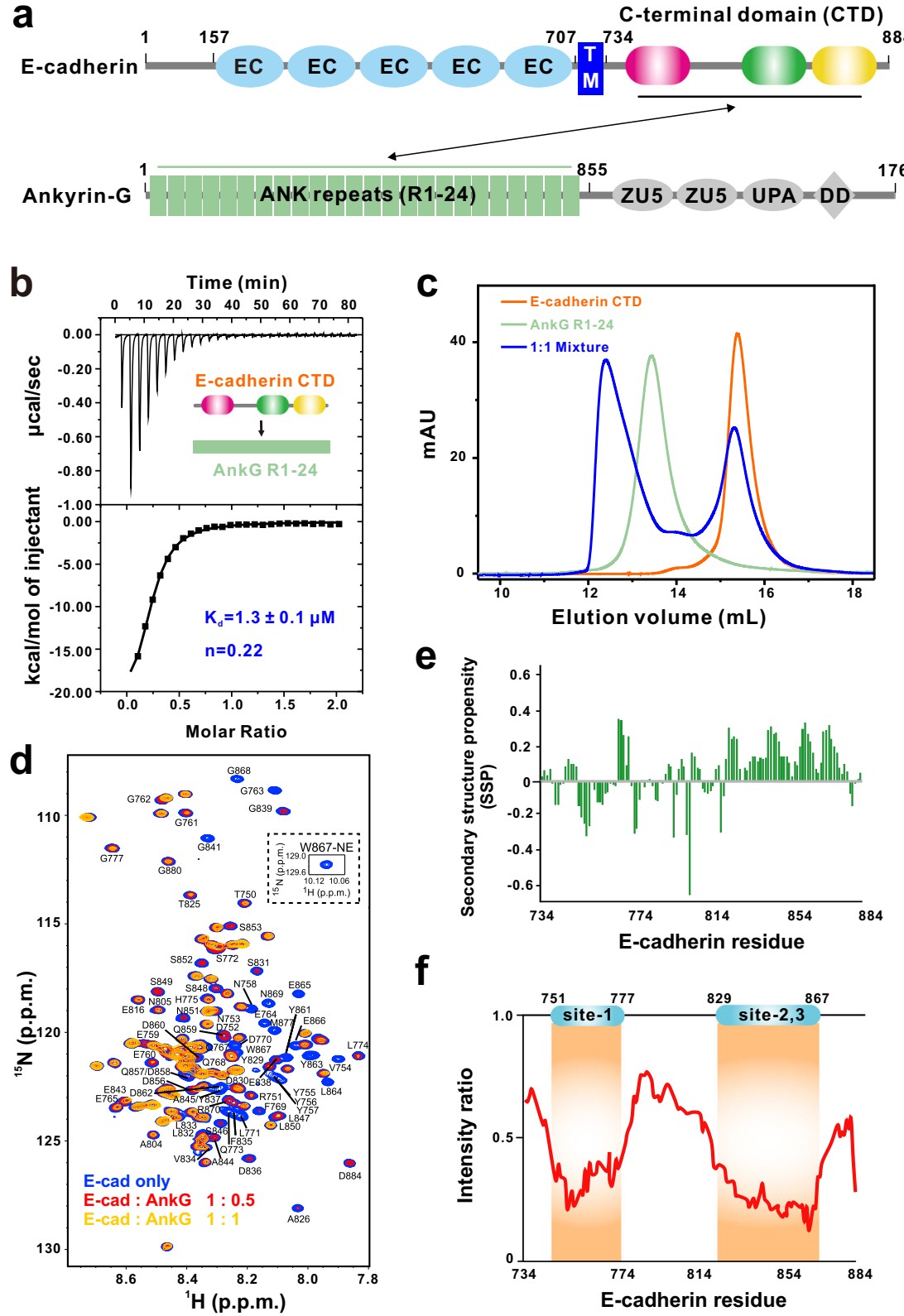

the E-cadherin tail into two parts (734-775, 776-884) based on previous literatures reporting its interaction with p120 catenin (734-775) or beta-catenin (776-884). We found both of these two fragments could bind to AnkG (Fig. S1). Subsequently we designed different truncations based on factors including conservation across species, amino acids properties, and the lengths of the fragments showed in Fig. S1. The ITC data support that there are three discrete sites within the E-cadherin

CTD that can interact with AnkG: 755-770 (site-1), 828-842 (site-2), and 854-869 (site-3), with $K_d$ values of ~0.9 μM, 1.9 μM, and 1.8 μM, respectively (Fig. S1 and Fig. 2a–c). This apparent multisite binding mode immediately sparked our interest in exploring E-cadherin-AnkG complex assembly, as this is distinct from previously known binding modes of both reported AnkG-Nav1.2 and AnkG-Neurofascin complexes and from E-cadherin-mediated complexes (e.g., E-cadherin-β-

**Fig. 1 | The E-cadherin C-terminal domain binds to AnkG R1-24. a** Schematic diagram showing the domain organization of E-cadherin and AnkG. EC, extracellular cadherin domains; TM, transmembrane domain; ANK repeats, ankyrin repeats; DD, death domain. **b** ITC-based measurement of the binding affinity of the E-cadherin CTD with AnkG R1-24. The $K_d$ error is the fitting error obtained using a 1-site binding kinetics model to fit the ITC data. **c** FPLC of E-cadherin CTD (residues 734-884, orange line), AnkG ANK repeats (R1-24, green line), and the 1:1 mixture of the complex (blue line). **d** Interactions assessed by [$^1$H,$^{15}$N]-HSQC NMR titrations of 100 μM $^{15}$N isotopically labeled E-cadherin CTD only (blue) or with two different concentrations of AnkG (red: 50 μM; yellow: 100 μM). **e** Secondary structure propensity (SSP) values of the E-cadherin CTD plotted as a function of the amino acid sequence. An SSP score at a given residue of 1 or −1 reflects a fully formed α-helical or β-sheet structure, respectively, whereas a score of zero indicates a completely unstructured conformation. **f** HSQC peak intensity ratios of E-cadherin CTD only (panel **d**, blue) to E-cadherin CTD with AnkG (panel **d**, yellow). The two regions in the E-cadherin CTD displaying dramatic intensity reductions are highlighted.

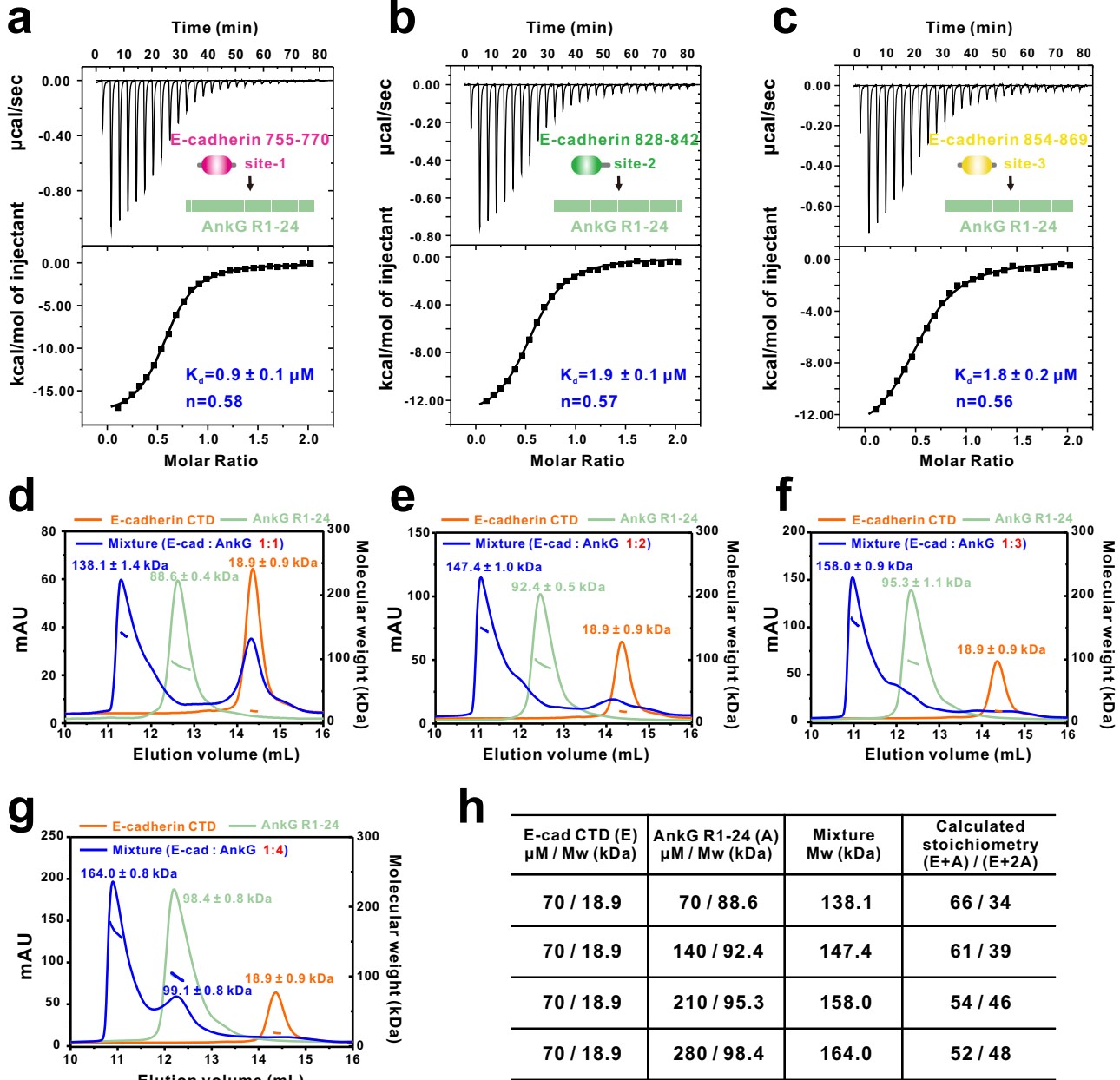

**Fig. 2 | Discrete sites within E-cadherin bind to AnkG in a dynamic manner.** ITC-based measurements of the binding affinities of AnkG R1-24 with E-cadherin residues 755-770 (site-1) (**a**), E-cadherin residues 828-842 (site-2) (**b**), and E-cadherin 854-869 (site-3) (**c**). **d**–**g** Analytical gel filtration chromatography coupled with static light scattering analysis of the E-cadherin CTD (orange), AnkG R1-24 (green), and E-cadherin CTD-AnkG R1-24 complex (blue) at the indicated molar ratios, showing that the E-cadherin CTD and AnkG R1-24 form an ensemble of heterogeneous complexes (specifically, the apparent stoichiometry indicates the presence of both a 1:1 complex and a 1:2 complex in solution). **h** Calculations based on the molecular weights of the complex peaks measured in panels **d**–**g**, indicating the distribution of the 1:1 and 1:2 E-cadherin CTD-AnkG R1-24 complexes.

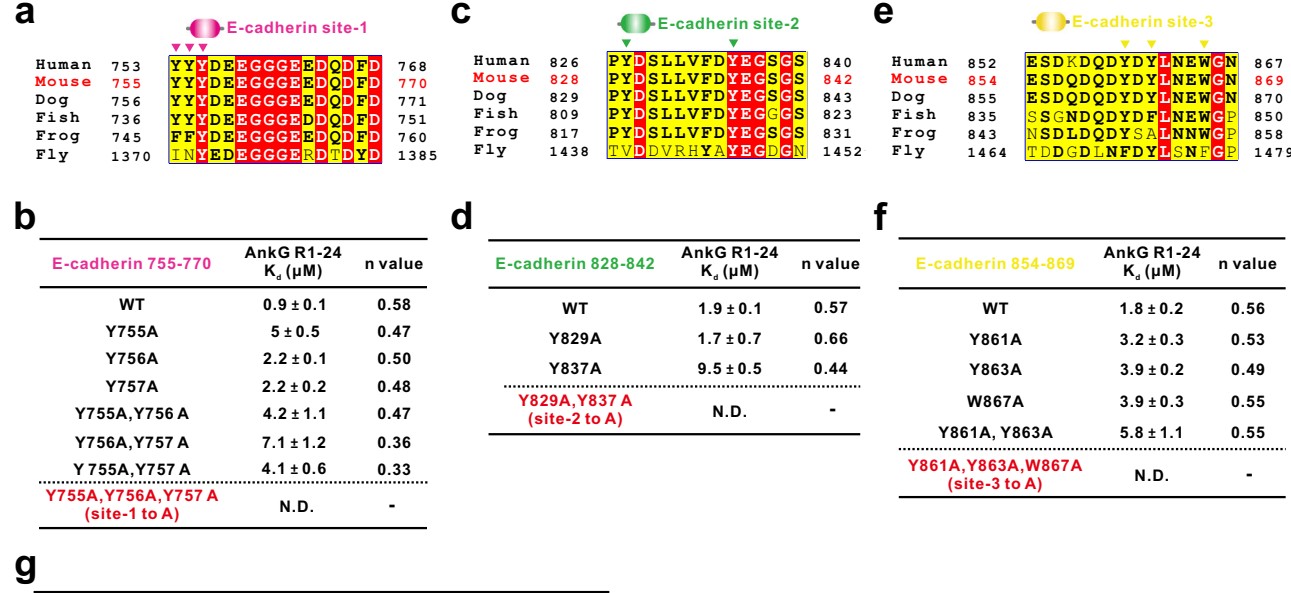

**Fig. 3 | Hydrophobic interactions involving aromatic residues of E-cadherin are responsible for binding with AnkG. a** Sequence alignment of E-cadherin site-1 in the indicated species, indicating strong conservation. Residues that are identical and highly similar are indicated in red and yellow boxes, respectively. **b** ITC binding assays showing the measured binding affinities between AnkG R1-24 and the indicated E-cadherin site-1 variants. N.D. indicates that no binding was detected. N.D. indicates that no binding was detected. **c** Sequence alignment of E-cadherin site-2. **d** ITC binding assays showing the measured binding affinities between AnkG R1-24 and the indicated E-cadherin site-2 variants. **e** Sequence alignment of E-cadherin site-3. **f** ITC binding assays showing the measured binding affinities between AnkG R1-24 and the indicated E-cadherin site-3 variants. **g** ITC-based measurement of binding affinities between AnkG R1–24 and the indicated variants of the E-cadherin CTD. The $n$ values and the fold changes of $K_d$ compared to the E-cadherin CTD WT are shown.

catenin, E-cadherin-p120 catenin), which exhibit strict 1:1 stoichiometry between two protein partners.

Specifically, we used AGFC coupled with a static light scattering assay to monitor the binding ratio between AnkG R1-24 and the E-cadherin CTD. When AnkG R1-24 and E-cadherin CTD were mixed at a 1:1 molar ratio, a complex peak was formed with a measured molecular weight of ~138.1 kDa. However, in addition to the complex peak, approximately 50% of the detected E-cadherin CTD was in an unbound form (Fig. 2d). The theoretical molecular weight of the complex comprising 1 copy of E-cadherin CTD and 1 copy of AnkG R1-24 is ~107.5 kDa, and that comprising 1 copy of E-cadherin CTD and 2 copies of AnkG R1-24 is ~196.1 kDa. Thus, it is clear that the detected complex peak (~138.1 kDa peak) was heterogeneous, with calculated proportions of 1:1 E-cadherin–AnkG complex (E + A) ~ 66% and 1:2 E-cadherin–AnkG complex (E + 2 A) ~34% (Fig. 2d, h).

When we further mixed the 2 components at 1:2, 1:3, and 1:4 molar ratios, the measured molecular weights of the complexes were ~147.4 kDa, 158.0 kDa, and 164.0 kDa, respectively, with the proportion of the 1:2 E-cadherin–AnkG complex increasing from ~39% to 48%, suggesting that the binding ratio of the complex tended to be stable but still in the range of 1:1 to 1:2, while the AnkG population was much more abundant than the E-cadherin CTD (Fig. 2e–h). Collectively, these results demonstrate that 1 E-cadherin molecule is able to bind to 1

AnkG molecule or to 2 AnkG molecules, forming a dynamic complex through discrete binding sites of E-cadherin.

To further confirm the binding stoichiometry between E-cadherin and AnkG, we evaluated whether a combination of the adjacent site-2 and site-3 of E-cadherin can bind to 2 AnkG molecules. ITC experiments using E-cadherin 828-884 (including both site-2 and site-3) and AnkG R1-24 displayed ~1.9 μM $K_d$ binding (Fig. S2a), which is comparable to either site-2 or site-3 alone in binding with AnkG R1-24. When we mixed AnkG R1-24 and E-cadherin 828-884 at a 1:1 molar ratio, a homogeneous complex peak was formed with a measured molecular mass of ~ 97.8 kDa, close to the theoretical molecular weight of a complex comprising 1 copy of E-cadherin 828-884 (~8.0 kDa) and 1 copy of AnkG R1-24 (~97.1 kDa) (Fig. S2b), suggesting that 1 E-cadherin 828-884 could only bind with 1 copy of AnkG R1-24. Thus, in our AGFC coupled with light scattering experiments, the binding stoichiometry for the E-cadherin-AnkG complex was determined to be between 1:1 and 1:2.

### Hydrophobic interactions involving aromatic residues of E-cadherin are responsible for binding with AnkG

We next conducted an amino acid sequence analysis of site-1, site-2, and site-3 of the E-cadherin CTD for AnkG binding. The results showed that these residues in each of the sites are conserved during evolution,

indicating that the binding mode between E-cadherin and AnkG is well preserved (Fig. 3a, c, and e). Based on the previous structural information of ANK repeat-mediated complex formation of ankyrins showing that hydrophobic interactions are necessary for the interaction[14–16], we speculated that the aromatic residues of E-cadherin may be required for binding with AnkG.

Thus, we evaluated the binding affinities between individual sites of E-cadherin variants bearing aromatic residue substitutions and AnkG R1-24 through ITC assays. We found that the bindings with AnkG R1-24 were completely abolished when we mutated several aromatic residues from each of the E-cadherin sites simultaneously, for example, Y755A, Y756A, and Y757A from site-1 (Fig. 3b), Y829A andY837A from site-2 (Fig. 3d), and Y861A, Y863A, and W867A from site-3 (Fig. 3f). However, single substitution of Y755, Y756, or Y757A from E-cadherin site-1 (Fig. 3b), Y829, or Y837 from E-cadherin site-2 (Fig. 3d), and Y861, Y863, or W867 from E-cadherin site-3 (Fig. 3f) to alanine only had slight effects on the binding with AnkG R1-24, decreasing the binding affinities only 2 - 5 times. Next, we further explored the binding affinities between full-length E-cadherin CTD variants and AnkG R1-24. ITC results showed that the E-cadherin CTD bearing single-site or double-site mutations still preserved considerable binding abilities with AnkG R1-24 ($K_d$ ranging from ~2.0 μM to ~6.3 μM), while only disrupting all three sites on the E-cadherin CTD (with a total of 8 aromatic residue substitutions) abolished complex formation (Fig. 3g). Taken together, these results indicate that for each of the binding sites of E-cadherin, the binding is dynamic, as it could tolerate any single aromatic residue substitution to retain the binding (Fig. 3b, d, f), and only by elimination of all three binding sites of E-cadherin could the interaction between E-cadherin and AnkG be blocked sufficiently (Fig. 3g).

## Promiscuous engagement of E-cadherin by AnkG

We next mapped the binding surface on AnkG for E-cadherin. We designed three truncations for the ANK repeat domain of AnkG, including ANK repeats 6-24 (R6-24), 8-24 (R8-24), and 8-14 (R8-14) (Fig. S3a–d). Through ITC analyses, we found that AnkG R8-24 is sufficient for binding with the E-cadherin CTD ($K_d$ ~1.5 μM) (Fig. S3d), E-cadherin 751-775 (site-1) ($K_d$ ~1.3 μM), and E-cadherin 828-869 (containing both site-2 and site-3) ($K_d$ ~2.5 μM), while AnkG R8-14 contains the core binding region for E-cadherin (Fig. S3b, c). These data indicated that AnkG used similar regions (AnkG R8-14) to bind to distinct E-cadherin binding sites (Fig. S3a–d).

To further investigate how the different binding sites of E-cadherin were engaged by AnkG, we first tried to crystallize the complex comprising AnkG and distinct binding sites of E-cadherin. Despite extensive efforts using all the ankyrin isoforms (AnkG, AnkR, and AnkB) of ankyrin family proteins in complex with E-cadherin binding sites, we failed to obtain high-resolution complex structures. We did obtain diffractive crystals for E-cadherin 828-842 fused with AnkR R6-16. However, we can only solve the rigid AnkR ANK repeat structure at 3.0 Å resolution, while the E-cadherin peptide densities were not sufficient to build an atomic model, indicating that the conformation of E-cadherin residues participating in binding is flexible (Fig. S4, Table S1).

Thus, we again turned to the NMR technique to overcome the obstacle of the dynamic nature of the E-cadherin-AnkG complex. In consideration of the molecular weight and behavior quality of proteins for NMR spectra, we took advantage of isotope labeling approaches and utilized the methyl-TROSY technique to assay complex formation[27,28]. There are a total of 14 methionine residues within AnkG R8-24, while 7 methionine residues are distributed evenly throughout the structure of AnkG R8-14 (Fig. 4a) and can serve as faithful NMR methyl probes to monitor binding. We collected $^1$H–$^{13}$C-correlated NMR spectra of AnkG R8-24 labeled in the methyl-bearing form of methionine. The resulting methyl-TROSY spectra are of high quality

(Fig. 4b). We then mutated the distinct 6 methionine residues (except the M291 located at R8) of AnkG R8-14 to alanine one by one and recorded the corresponding spectrum of the AnkG R8-24 variants to achieve the methionine assignments (Fig. 4b and Fig. S5a–f).

To determine the exact E-cadherin binding sites on AnkG, we made fusion proteins that linked the entire E-cadherin CTD, as well as each individual binding site, to the N-terminus of AnkG R8-24 to trap the fast dissociating reactions. We found that upon E-cadherin binding, the NMR signals of AnkG experienced drastic attenuation accompanied by chemical shift changes, featuring a highly dynamic interaction process (Fig. 4c–f), consistent with the biochemical results shown above (Figs. 1 and 2). Mapping of the residues including M348, M451, and M480 that undergo dramatic NMR signal perturbations upon E-cadherin binding allows us to define a shallow pocket on AnkG with a size of ~20 × 45 Å, spanning repeats 8 to 14 (Fig. 4a, c–f). Distinct from previously reported targets for ANK repeats of ankyrins, charge–charge interactions merely contribute to the binding (Fig. S6a, b). Regardless of distinct amino acid compositions, all individual E-cadherin binding sites (Fig. 4d–f), as well as the whole CTD of E-cadherin (Fig. 4c), exerted binding effects, albeit to different extents, to essentially identical regions on AnkG, indicative of a conserved binding pocket that AnkG utilizes to promiscuously engage multiple E-cadherin binding sites.

Comparison of the CSP patterns, an exquisitely sensitive probe representing binding manners, upon binding to individual E-cadherin sites and the whole CTD of E-cadherin reveals that the CSP profile upon binding to the E-cadherin CTD is approximately an average of that of three segments (Fig. 4g). This result suggests that the three binding sites on the E-cadherin CTD, when occurring simultaneously with AnkG, compete and exchange dynamically at the AnkG binding pocket on a fast-exchange NMR time scale. Taken together, the data demonstrate that the interaction of E-cadherin and AnkG is highly dynamic, as also evidenced by our ITC data (Figs. 1b and 2a–c).

This dynamic interaction mechanism is consistent with the size of the binding pocket on AnkG identified here, which is sufficient for swaddling only a single individual E-cadherin binding segment but not multiple sites at the same time. In this scenario, multiple E-cadherin binding sites dynamically compete for the limited space provided by AnkG and only transiently stay, and at any given time, multiple binding conformers coexist, resulting in a "fuzzy" protein complex (Fig. 4h). Indeed, the intrinsic structural heterogeneity of the E-cadherin-AnkG complex is consistent with our crystallography data that densities of E-cadherin peptides could not be traced well due to dynamic equilibrium (Fig. S4).

## Effects of the aromatic residue mutations from E-cadherin on binding with β-catenin or p120 catenin

Before investigating the functional consequences of the interaction between E-cadherin and AnkG in the cellular contexts, we also evaluated the effects of our aforementioned E-cadherin CTD variants bearing different aromatic residue substitutions (7 variants in total) (Fig. 3g) on binding with β-catenin and p120 catenin using ITC assays (Fig. 5a, b). E-cadherin site-2 to A or other constructs bearing these site mutations apparently affect β-catenin binding to E-cadherin (Fig. 5a), while site-1 of E-cadherin is responsible for p120 catenin binding (Fig. 5b), consistent with previously defined binding modes for canonical cadherin-catenin complexes.

## E-cadherin variants bearing AnkG-binding deficient mutations alter accumulation at lateral membranes

Previous studies reported that accumulation of E-cadherin at the lateral membrane in both epithelial cells and early embryos required binding with AnkG[22], and knockdown of AnkG blocks E-cadherin localization at sites of cell–cell contact. To functionally dissect the E-cadherin-AnkG complex, we used polarized MDCK cells to monitor the

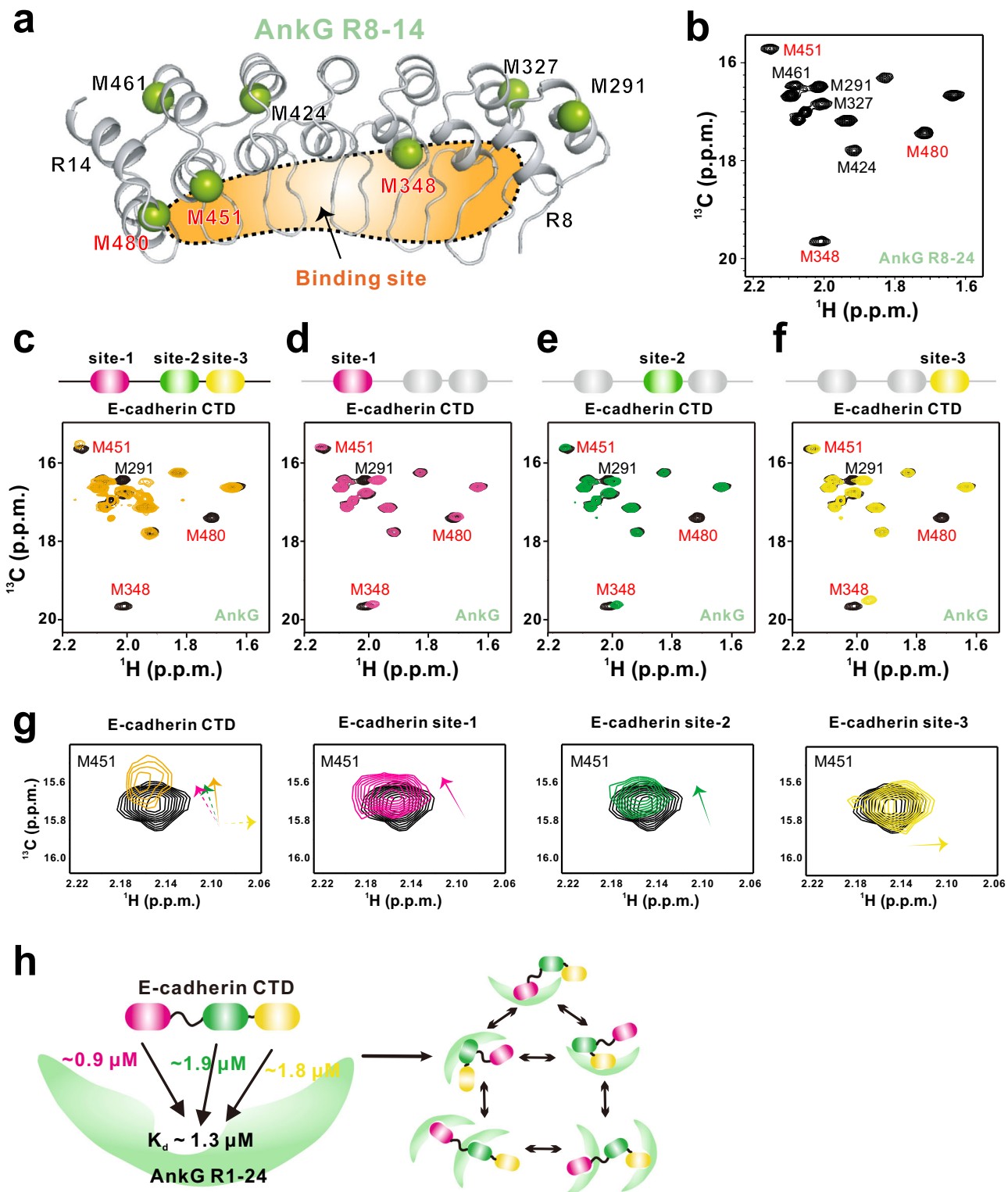

presence of both E-cadherin and AnkG at lateral membranes. Briefly, we constructed wild-type (WT) GFP-E-cadherin and seven GFP-E-cadherin variants bearing aforementioned aromatic residue substitutions (the E-cadherin vectors we used here are from human species), including Y753A, Y754A, and Y755A triple mutations (site-1 to A), Y827A and Y835A double mutations (site-2 to A), Y859A, Y861A, and W865A triple mutations (site-3 to A), and combinations of these site-1, site-2, and site-3 variants (site-1, 2 to A, site-1, 3 to A, site-2, 3 to A, and site-1, 2, 3 to A). When these constructs were overexpressed in MDCK cells, the

WT GFP-E-cadherin exclusively accumulated at the lateral membranes, while the site-1, 2, 3 to A variant exhibited no lateral membrane localization at all (Fig. S7a–i). Some of the variants, including site-1 to A, site-1, 2 to A, site-1, 3 to A, and site-2, 3 to A, showed a slight cytosolic mislocalization phenotype but still retained membrane accumulation (Fig. S7b, e–g, and i).

We next generated a stable cell line expressing a shRNA targeting E-cadherin, which caused a more than 80% reduction in E-cadherin levels compared with the empty vector control (Fig. S8a, b). We

**Fig. 4 | Promiscuous engagement of E-cadherin by AnkG revealed by NMR.**
**a** Ribbon-stick diagram showing the relatively uniform distribution of the 7 Met residues in the AnkG R8-14 structure (the figure is generated using our reported structure with PDB code 7XCE). The site of AnkG binding to the E-cadherin CTD is indicated in orange. The figure of AnkG R8-14 was generated using complex structure of AnkG-Neurofasin (PDB code 7XCE). **b** [$^{1}$H–$^{13}$C]-correlated methyl-TROSY spectra of Met-$^{13}$CH$_3$ isotopically labeled AnkG R8-24 (with assignments indicated). Overlay of [$^{1}$H–$^{13}$C]-correlated methyl-TROSY spectra of Met-$^{13}$CH$_3$-labeled AnkG R8-24 (black) and AnkG R8-24 fusion with the E-cadherin CTD (**c**; orange), E-cadherin site-1 (**d**; pink), E-cadherin site-2 (**e**; green), and E-cadherin site-3 (**f**; yellow). The fact that the same AnkG residues (M348, M451, and M480) display significant interaction effects upon binding to the E-cadherin CTD or individual sites 1–3 is indicative of a promiscuous binding mechanism. **g** CSP pattern analysis of M451 from AnkG R8-24 binding to the E-cadherin CTD or individual E-cadherin sites 1-3. The CSP directions are indicated by arrows. The CSP of M451 upon binding to the E-cadherin CTD displays an approximate averaging effect of the three discrete sites, indicative of dynamic exchange. **h** A model depicting the dynamic interactions between the E-cadherin CTD and AnkG R1-24 in which three discrete binding sites from E-cadherin compete with the same binding region of AnkG to form a promiscuous complex. The E-cadherin CTD could form a 1:1 complex with AnkG through the individual of the three binding sites, and it can also form a 1:2 complex with AnkG through site-1 with site-2 or site-3, probably due to the steric hinderance of the adjacent site-2 and site-3.

**a**

| E-cadherin 734-884 (CTD) | β-catenin 1-781 $K_d$ (μM) | n value | fold change |
|---|---|---|---|
| WT | 0.036 ± 0.004 | 0.80 | 1 |
| site-1 to A | 0.056 ± 0.004 | 0.98 | ~1.5 |
| site-2 to A | 4.8 ± 0.6 | 0.76 | >100 |
| site-3 to A | 0.11 ± 0.01 | 0.85 | ~3.1 |
| site-1,2 to A | 3.6 ± 0.3 | 0.77 | >100 |
| site-1,3 to A | 0.098 ± 0.004 | 0.72 | ~2.7 |
| site-2,3 to A | 1.4 ± 0.2 | 0.81 | ~38 |
| site-1,2,3 to A | 8.1 ± 0.7 | 0.74 | >200 |

**b**

| E-cadherin 734-884 (CTD) | p120 catenin 324-938 $K_d$ (μM) | n value | fold change |
|---|---|---|---|
| WT | 1.7 ± 0.1 | 0.92 | 1 |
| site-1 to A | >50 | - | >30 |
| site-2 to A | 2.8 ± 0.1 | 0.5 | ~2 |
| site-3 to A | 2.9 ± 0.1 | 0.84 | ~2 |
| site-1,2 to A | >70 | - | >40 |
| site-1,3 to A | >30 | - | ~20 |
| site-2,3 to A | 1.8 ± 0.1 | 0.8 | ~1 |
| site-1,2,3 to A | >50 | - | >30 |

**Fig. 5 | Effects of the aromatic residue mutations from E-cadherin on binding with β-catenin or p120 catenin. a** ITC binding assays showing the measured binding affinities between β-catenin 1-781 and the indicated E-cadherin CTD WT and variants. **b** ITC binding assays showing the measured binding affinities between p120 catenin 324-938 and the indicated E-cadherin CTD WT and variants. The n values and the fold changes of $K_d$ compared to the E-cadherin CTD WT are shown.

transfected the aforementioned GFP-E-cadherin WT and variant constructs into the stable cell line to rescue E-cadherin expression and stained the E-cadherin, AnkG, and actin signals (Fig. 6a–h). We observed similar phenomena for E-cadherin signals: only site-1, 2, 3 to A showed altered lateral membrane localization (Fig. 6h), while other variants still preserved membrane accumulation with some extent missorting to the cytosol (Fig. 6b–g, and i). The cytosol/membrane ratios of E-cadherin intensities for double site variants were clearly higher than those of single site variants (Fig. 6i). The endogenous AnkG signals showed a similar pattern to that of E-cadherin, indicating functional complex formation at the lateral membranes (Fig. 6a–h, and j). Not surprisingly, the E-cadherin site-1, 2, 3 to A, which completely lost membrane localization, also failed to recruit AnkG to the lateral membranes (Fig. 6h). Collectively, these results demonstrated that the correct accumulation of E-cadherin at lateral membranes requires AnkG binding.

We further took advance of our stable cell lines to examine whether the localizations of canonical catenin proteins, including β-catenin, α-catenin and p120-catenin, were affected when E-cadherin was knocked down. We stained the endogenous β-catenin, α-catenin and p120-catenin signals in the stable cell lines rescued with WT E-cadherin, control vector (GFP control), and our aforementioned E-cadherin variants (Fig. 7a–r). It is worth noting that the differences in cell areas in the GFP control and some of the E-cadherin mutants are due to the lack of polarization (Fig. 7h, i, q, and r). The results showed that there were no obvious effects on β-catenin, α-catenin, or p120 catenin localizations and they remained the membrane accumulation in cells depleting E-cadherin (Fig. 7a–u), showing distinct behaviors compared with AnkG's localization. One of the potential mechanisms may be that MDCK cells express several other classical cadherins, e.g. cadherin-6 (K-cadherin), a type II atypical cadherin. When we knocked down E-cadherin, these catenin proteins are stabilized by these classical cadherins, especially the K-cadherin. In contrast, AnkG mislocalized in cytosol when depleting E-cadherin, presenting the different binding mechanisms described in vitro may contribute the different cellular phenomenon.

## E-cadherin-AnkG complex functions in the organization of lateral membranes

We also evaluated the organization of lateral membranes in MDCK cells, a characteristic of epithelial cell polarity. In the E-cadherin knockdown stable cell line transfected with GFP control vector, the cells exhibited slow growth, and the lateral membranes collapsed with a very limited height (Fig. S8c). When rescued with WT E-cadherin, the cells could restore a normal polarized phenotype and reached an average height of ~8.5 μm within 72 h (Fig. 6k, l). In contrast, the lateral membrane height of cells rescued with E-cadherin site-1,2,3 to A variant or GFP control vector showed a marked reduction that remained at approximately 5–6 μm after 72 h in culture (Fig. 6k, l). When rescued with E-cadherin variants bearing other site mutations, the height of lateral membranes could also be restored (Fig. 6k, l). These results demonstrate that the assembly of the E-cadherin–AnkG complex functions directly in the organization of lateral membranes.

## Orchestration of E-cadherin by AnkG and β-catenin simultaneously

Next, we wondered why E-cadherin adopts a unique interaction mode to bind with AnkG compared with the previously known E-cadherin-catenin complex and sought to determine the relationships between the E-cadherin-AnkG complex and the E-cadherin-catenin complex. We used AGFC coupled with static light scattering assays to investigate the connections between these protein complexes. We prepared AnkG R1-24, E-cadherin CTD, and full-length β-catenin 1-781 proteins individually, and the column behaviors of these proteins appeared sufficient to

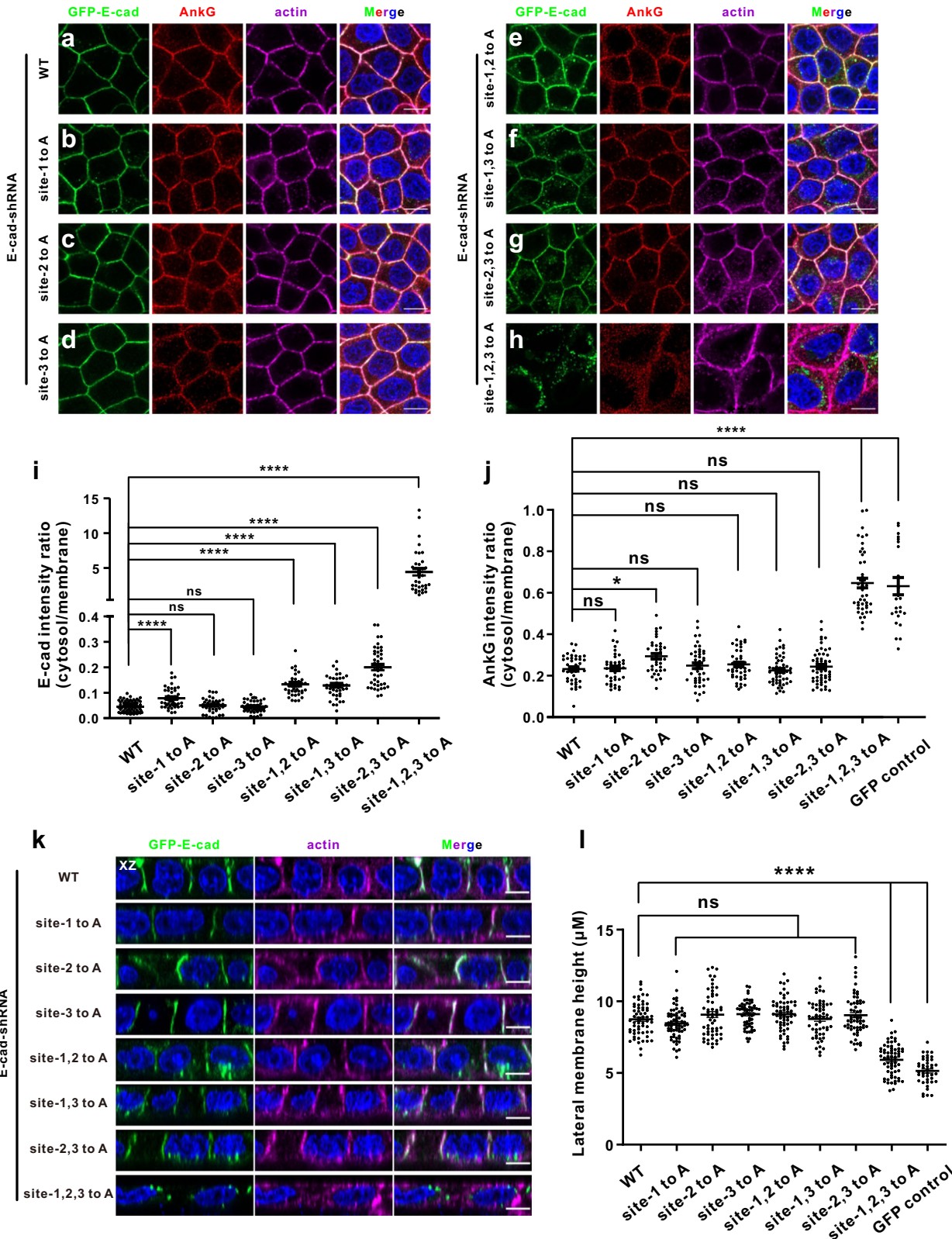

enable evaluation of complex formation upon mixing of these proteins. When AnkG R1-24, E-cadherin CTD, and β-catenin were mixed at 1:1:1, 2:1:1, 3:1:1, and 4:1:1 molar ratios, complex peaks were formed with respective measured molecular weights of ~225.4 kDa, 245.5 kDa, 268.5 kDa, and 266.3 kDa (Fig. 8a), indicating that AnkG R1-24, E-cadherin CTD, and β-catenin can form a three-subunit complex with apparent binding ratios ranging between 1:1:1 and 2:1:1. The three-subunit complex was also confirmed by ITC experiments using titrations of the E-cadherin CTD−AnkG R1-24 complex or E-cadherin 776-884−AnkG R1-24 complex to β-catenin (Fig. 8b & Fig. S9a). Titration of the E-cadherin CTD−β-catenin complex to AnkG R1-24 also showed interactions, further supporting the formation of a three-subunit complex (Fig. S9b, c). Collectively, these results demonstrate that AnkG and β-catenin can coordinately orchestrate the E-cadherin

**Fig. 6 | The E-cadherin–AnkG complex is essential for the organization of lateral membranes.** MDCK cells stably expressing a shRNA targeting E-cadherin were transfected with wild-type (WT) GFP-E-cadherin (**a**), GFP-E-cadherin site-1 to A (**b**), GFP-E-cadherin site-2 to A (**c**), GFP-E-cadherin site-3 to A (**d**), GFP-E-cadherin site-1, 2 to A (**e**), GFP-E-cadherin site-1, 3 to A (**f**), GFP-E-cadherin site-2, 3 to A (**g**), and GFP-E-cadherin site-1, 2, 3 to A (**h**). The cells were stained with GFP signals showing E-cadherin localization, endogenous AnkG signals showing AnkG localization, and actin signals showing the lateral membranes. Scale bars: 10 μm. Quantification of the immunofluorescence intensity ratios of cytosolic to membranes for E-cadherin WT ($n = 70$); site-1 to A ($n = 45$); site-2 to A ($n = 48$); site-3 to A ($n = 52$); site-1, 2 to A ($n = 39$); site-1, 3 to A ($n = 41$); site-2, 3 to A ($n = 42$); and site-1, 2, 3 to A ($n = 35$) (**i**); or for AnkG in the groups of WT ($n = 41$); site-1 to A ($n = 40$); site-2 to A ($n = 37$); site-3 to A ($n = 44$); site-1, 2 to A ($n = 43$); site-1, 3 to A ($n = 53$); site-2, 3 to A ($n = 55$); site-1, 2, 3 to A ($n = 43$); and GFP control ($n = 25$) (**j**). Data are presented as the means ± SEMs and analyzed using one-way ANOVA followed by Dunnett's multiple comparisons test to WT E-cadherin (**i**) or AnkG (**j**), *$p = 0.0194$, ****$p < 0.0001$, ns, not significant. (**k**) XZ projections of panels **a**–**h**, showing heights of the lateral membranes rescued with various GFP-E-cadherin constructs as indicated. Scale bars: 5 μm. **l** Quantification of lateral membrane heights labeled by the immunofluorescence intensity of the actin signals from panel **k** in the groups of WT ($n = 62$); site-1 to A ($n = 68$); site-2 to A ($n = 56$); site-3 to A ($n = 57$); site-1, 2 to A ($n = 61$); site-1, 3 to A ($n = 58$); site-2, 3 to A ($n = 61$); site-1, 2, 3 to A ($n = 65$); and GFP control ($n = 43$). Data are presented as means ± SEM and analyzed using one way ANOVA followed by Dunnett's multiple comparisons test to WT. ****$p < 0.0001$, ns, not significant.

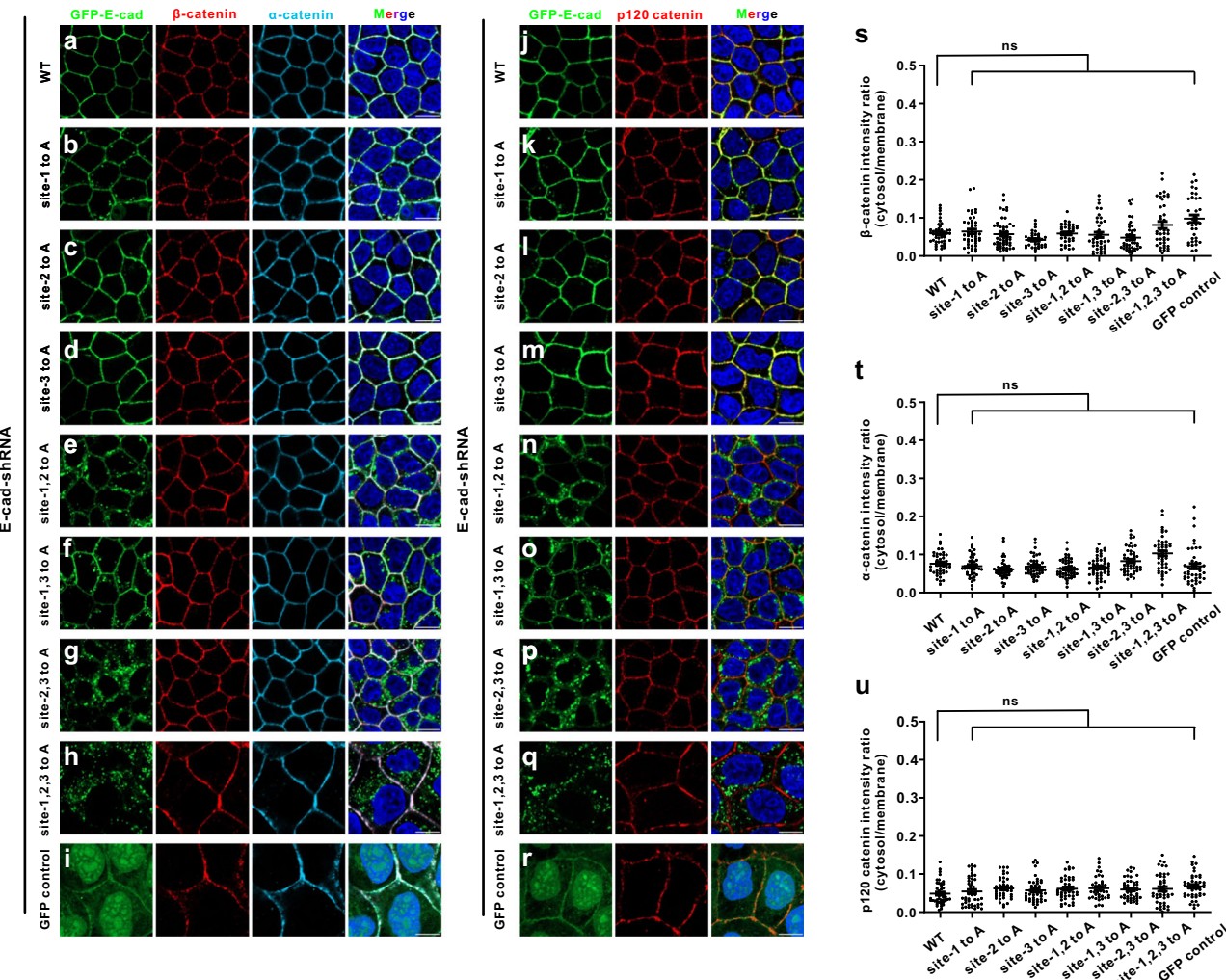

**Fig. 7 | Catenin proteins preserve membrane localizations when E-cadherin is knocked down in MDCK cells.** MDCK cells stably expressing a shRNA targeting were transfected with wild-type (WT) GFP-E-cadherin (**a**), GFP-E-cadherin site-1 to A (**b**), GFP-E-cadherin site-2 to A (**c**), GFP-E-cadherin site-3 to A (**d**), GFP-E-cadherin site-1, 2 to A (**e**), GFP-E-cadherin site-1, 3 to A (**f**), GFP-E-cadherin site-2, 3 to A (**g**), GFP-E-cadherin site-1, 2, 3 to A (**h**), and GFP vector (**i**) respectively. The cells were stained with GFP signals showing E-cadherin localization, endogenous β-catenin and α-catenin signals showing β-catenin and α-catenin localization (Scale bars: 10 μm.). MDCK cells stably expressing a shRNA targeting were transfected with WT GFP-E-cadherin (**j**), GFP-E-cadherin variants (**k**–**q**), and GFP vector (**r**) respectively. Endogenous p120 catenin signal showing p120 catenin localization (Scale bars: 10 μm.) (**s**) Quantification of the immunofluorescence intensity ratios of cytosolic β-catenin *vs* plasma membranes for E-cadherin WT ($n = 46$); site 1 to A ($n = 43$); site 2 to A ($n = 50$); site 3 to A ($n = 35$); site 1, 2 to A ($n = 37$); site 1, 3 to A ($n = 40$); site 2, 3 to A ($n = 44$); site 1, 2, 3 to A ($n = 41$); GFP vector ($n = 36$). **t** Quantification of the immunofluorescence intensity ratios of cytosolic α-catenin versus plasma membranes for E-cadherin WT ($n = 44$); site 1 to A ($n = 43$); site 2 to A ($n = 44$); site 3 to A ($n = 46$); site 1, 2 to A ($n = 50$); site 1, 3 to A ($n = 50$); site 2, 3 to A ($n = 48$); site 1, 2, 3 to A ($n = 46$); GFP vector ($n = 47$). **u** Quantification of the immunofluorescence intensity ratios of cytosolic p120 catenin versus plasma membranes for E-cadherin WT ($n = 47$); site 1 to A ($n = 47$); site 2 to A ($n = 42$); site 3 to A ($n = 42$); site 1, 2 to A ($n = 43$); site 1, 3 to A ($n = 38$); site 2, 3 to A ($n = 47$); site 1, 2, 3 to A ($n = 43$); GFP vector ($n = 44$). Data are presented as means ± SEM and analyzed using one-way ANOVA followed by Dunnett's multiple comparisons test to β-catenin (**s**), α-catenin (**t**), and p120 catenin (**u**). ns, not significant.

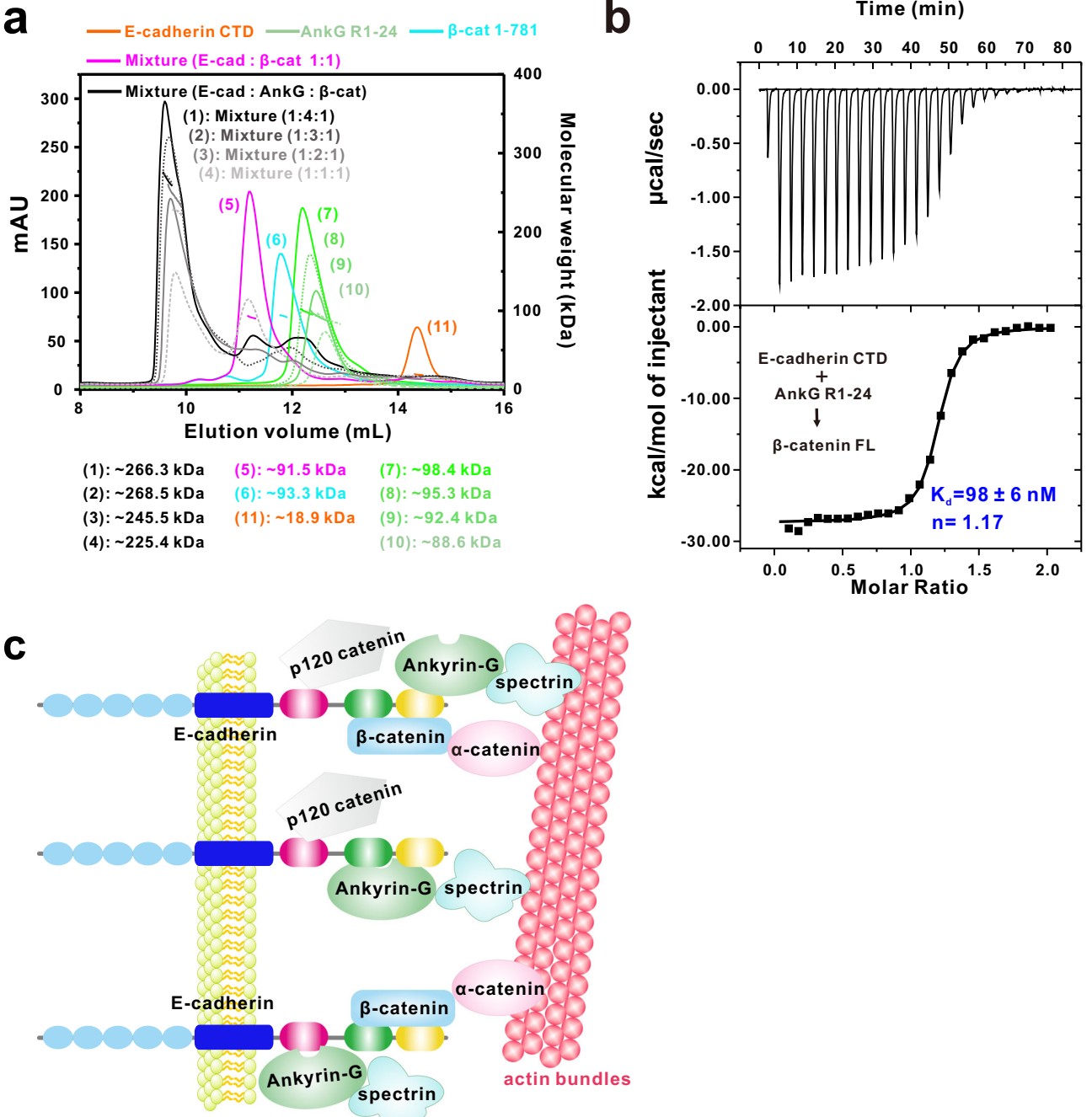

**Fig. 8 | Orchestration of E-cadherin by AnkG and β-catenin simultaneously.**
**a** Analytical gel filtration chromatography coupled with static light scattering analysis of the E-cadherin CTD (orange), AnkG R1-24 (green, four concentrations), β-catenin 1-781 (cyan), E-cadherin CTD–β-catenin 1-781 complex (magenta), and E-cadherin CTD–AnkG R1-24–β-catenin triple complex at the indicated molar ratios (black), showing that E-cadherin CTD, AnkG R1-24, and β-catenin can form a three-subunit complex. **b** ITC-based measurement of the binding affinity of the E-cadherin CTD-AnkG R1-24 complex with β-catenin FL (full length). **c** Schematic model of the E-cadherin-mediated protein complexes in lateral membranes. p120 catenin interacts with the N-terminus of the E-cadherin CTD. β-catenin binds to the C-terminal ~100 residues of the E-cadherin CTD. Three discrete binding sites of E-cadherin mediate dynamic interactions with AnkG, ensuring that E-cadherin is tethered to AnkG (linking to actin bundles via spectrin) regardless of other target binding status.

complex and AnkG can consistently be tethered to the E-cadherin tail through the dynamic nature of the binding between AnkG and E-cadherin (Fig. 8c).

## Discussion

The cell adhesion molecule E-cadherin is required for the formation of lateral membranes and the maintenance of cell polarity and is a known tumor suppressor[12,29,30]. Loss of E-cadherin expression is a hallmark of the epithelial-mesenchymal transition process. Therefore, elucidation

of the molecular mechanisms governing E-cadherin-mediated complex assembly can advance the understanding of the establishment and maintenance of epithelial cell polarity. In this study, we characterized the dynamic interactions between E-cadherin and AnkG. Extensive biochemical and biophysical analyses demonstrated that three discrete binding sites of E-cadherin mediate interactions with the same region on AnkG ANK repeats. The binding affinity for the entire E-cadherin CTD and AnkG was comparable to or even slightly weaker than the individual binding sites of the E-cadherin CTD (~1.3 μM *vs*

0.9 μM, 1.9 μM, and 1.8 μM, respectively). These results demonstrate no apparent binding avidity of the E-cadherin sites and AnkG, probably because the linker tethering the (sub)sites remains dynamic even in the complex with AnkG, and the resulting entropy penalty counteracts the synergistic effect, thus allowing for quasi-independent binding of the individual E-cadherin sites.

Distinct from the canonical E-cadherin-β-catenin and E-cadherin-p120 catenin complexes, which form a typical 1:1 homogeneous complex, the E-cadherin-AnkG complex engages through a distinct interaction mode: multiple E-cadherin binding sites competitively interact with the limited space provided by the same pocket of AnkG, resulting in a "fuzzy" protein complex (Fig. 4h). On the one hand, this dynamic interaction mechanism is consistent with the size of the binding pocket on AnkG identified here (AnkG R8-14), which is sufficient for swaddling ~15 amino acids, i.e., a single individual E-cadherin binding segment, but not multiple sites at the same time. On the other hand, although there are three discrete binding sites present in the E-cadherin CTD, the binding stoichiometry with AnkG is restricted to 1:1 or 1:2, probably due to the steric hindrance for the adjacent site-2 and site-3 of the E-cadherin CTD.

In our previous studies, we solved the crystal structures of ANK repeats in complex with their own autoinhibitory peptides and with Nav1.2 and demonstrated that both hydrophobic and electrostatic interactions mediate interactions between ANK repeats and their targets[15,16]. However, we only identified essential aromatic residues of the E-cadherin CTD that are responsible for binding with AnkG, showing that hydrophobic interactions are necessary. High salt concentrations did not obviously alter the binding between AnkG and E-cadherin, indicating a previously unrecognized binding mode for ANK repeat of ankyrins in which charge–charge interactions are dispensable. We think this is another level for the dynamic nature of the E-cadherin-AnkG complex: in addition to the three binding sites of E-cadherin dynamically competing for the binding pocket on AnkG, each individual binding site of E-cadherin itself also dynamically interacts with AnkG mainly through hydrophobic interactions that lack specificity. Mechanistically, this dynamic interaction mode substantially expands our understanding of both the AnkG ANK repeat-mediated complex and the E-cadherin complex but is reminiscent of the chaperone-like behavior that dynamically attaches to client proteins to ensure persistent complex assembly[31,32].

Functionally, as a scaffold protein that links diverse membrane proteins to the underlying spectrin-based cytoskeleton, AnkG has been reported to bind with E-cadherin and is required for the restricted mobility of E-cadherin and the formation of lateral membranes[10,22]. Here, we demonstrated that AnkG stabilized the localization of E-cadherin on the lateral membranes and showed that disruption of the binding between E-cadherin and AnkG led to mislocalization of E-cadherin and a reduction in lateral membrane height (Fig. 6a–l). Thus, AnkG binding with E-cadherin is essential for the organization of lateral membranes. We think that the dynamic interactions between E-cadherin and AnkG identified in this study ensure that E-cadherin is tethered to AnkG under various cellular conditions. Notably, Ankyrin is reported to regulate trafficking of several membrane proteins including Na$^+$/K$^+$ ATPase, suggesting that Ankyrin functions not only on the maintenance of membrane protein localization, but also on the establishment of the membrane protein polarity. There is no direct evidence that AnkG regulates E-cadherin trafficking, but it is possible that AnkG participates in the process as the dynamic nature of the binding may ensure the complex formation.

The E-cadherin-catenin complex is well known for the establishment of lateral membranes. We biochemically found that E-cadherin, AnkG, and β-catenin could form a three-subunit complex mediated by the E-cadherin CTD, indicating that the E-cadherin–AnkG and E-cadherin–β-catenin complexes work together to guarantee the integrity of lateral membranes. β-catenin is not only a molecule of cell adhesion but also plays essential roles in the Wnt signaling cascade[33]. Interestingly, a previous study reported that AnkG could regulate neurogenesis by altering the subcellular localization of β-catenin by stabilizing the cadherin-catenin complex in the nervous system[34], beautifully supporting our findings that AnkG cooperates with the E-cadherin–β-catenin complex. Again, this dynamic interaction mode fits well with the physiological functions of E-cadherin and the AnkG complex in cell polarity maintenance, and the E-cadherin–AnkG–β-spectrin pathway may be the principal pathway for the connection of E-cadherin with the cytoskeleton.

Taken together, we have illustrated the relationships between E-cadherin–AnkG and the E-cadherin–β-catenin complex and dissected the detailed dynamic interactions of E-cadherin and AnkG involved in epithelial cell polarity maintenance through a combination of multiple biological methods. These findings not only provide insights into the organization of multiple E-cadherin-mediated complexes and a structural basis for further investigation of these disease-related protein complexes but also may shed light on potential treatments and drug discoveries for related human diseases.

## Methods

### Protein expression and purification

The full-length coding sequences of E-cadherin ("UniProt: P09803"), β-catenin ("UniProt: Q02248"), and p120 catenin ("UniProt: P30999") were PCR amplified from a mouse cDNA library. Various mutations or shorter fragments of E-cadherin and AnkG were generated using standard PCR-based methods and confirmed by DNA sequencing. The primer sequences used in this study were listed in the Supplementary Table 2. The coding sequences of various proteins were cloned into a home-modified pET32a vector with an N-terminal thioredoxin (Trx)-His$_6$-tag. All the constructs were expressed in Escherichia coli BL21 (DE3) host cells at 16 °C for 20 hours and induced by the addition of 0.2 mM isopropyl-β-D-thiogalactoside (IPTG). Cells were harvested and purified by Ni$^{2+}$-NTA agarose affinity columns followed by size-exclusion chromatography (Superdex 200 or Superdex 75) with buffer containing 50 mM Tris, 100 mM NaCl, 1 mM DTT and 1 mM EDTA at pH 7.8. The Trx-His$_6$-tag was removed by incubation with HRV 3 C protease and separated by size exclusion columns when needed.

### Isothermal titration calorimetry (ITC) assay

ITC measurements were carried out on a Microcal VP-ITC calorimeter (Malvern) at 25 °C. Proteins used for ITC measurements were dissolved in an assay buffer composed of 50 mM Tris, 100 mM NaCl, 1 mM EDTA, and 1 mM DTT at pH 7.8. High concentrations of proteins (e.g., 200 μM for Trx-E-cadherin) were individually loaded into the syringe and titrated into the cell containing low concentrations of the corresponding proteins (e.g., 20 μM for Trx-AnkG R1-24). For each titration point, a 10 μl aliquot of a protein sample in the syringe was injected into the protein in the cell at a time interval of 120-80 seconds. ITC titration data were analyzed using Origin7.0 software and fitted by the one-site binding model. The n values for all ITC experiments are listed in the figure or summary table.

### Fast protein liquid chromatography coupled with static light scattering

The analysis was performed on an AKTA FPLC system (GE Healthcare) coupled with a static light scattering detector (miniDawn, Wyatt) and a differential refractive index detector (Optilab, Wyatt). Protein samples (70 μM for E-cadherin and 70 μM or 140 μM or 210 μM or 280 μM for AnkG) were filtered and loaded into a Superdex 200 increase column pre-equilibrated by a column buffer composed of 50 mM Tris, 100 mM NaCl, 1 mM EDTA, and 1 mM DTT at pH 7.8. Data were analyzed with ASTRA6 (Wyatt).

## Cell culture

MDCK and HEK293 cells were obtained from the American Type Culture Collection and maintained in a humidified environment at 37 °C with 5% $CO_2$. Cells were cultured in DMEM (Gibco with 10% fetal bovine serum, 100 units/ml penicillin, and 100 units/ml streptomycin).

## shRNA and lentiviral infection

For endogenous gene knockdown experiments, shRNA targeting dog E-cadherin with the sequence 5'-GCAGCATGATGTTCACTATCA-3' was inserted into the lentivirus-based PLKO.1 vector via EcoRI and AgeI sites. For E-cadherin rescue experiments, the E-cadherin WT or variants was inserted into the lentivirus-based pLVX-EGFP vector. PLKO.1, psPAX2, and pMD2.G were ordered from Addgene. Lentivirus packaging was performed by using Lipo 3000-based cotransfection of HEK293 cells with psPAX2, pMD2.G, and the lentiviral vector PLKO.1 (for shRNA) or pLVX-EGFP (for E-cadherin). Supernatant medium of packaging cells was harvested up to 48 h after transfection and filtered through a 0.22-μm filter. For infection, MDCK cells with 50–60% confluency were cultured in DMEM and mixed with virus and 8 μg/ml polybrene (Sigma). After incubation for 12 h, the culture medium was changed to DMEM containing 10% FBS followed by selection with puromycin.

## Antibodies and immunofluorescence imaging

Rabbit anti-E-cadherin (Proteintech 20874-1-AP; 1:1000 for western blotting (WB); 1:500 for immunofluorescence (IF)); anti-AnkG (4G3F8; Life Technologies, 1:500 for IF); Alexa Fluor 647-Phalloidin (Invitrogen, A22287, 1:1000 for IF); Rabbit anti-GAPDH (Proteintech, 10494-1-AP, 1:2000 for WB); Rabbit anti-p120 catenin (Proteintech, 12180-1-AP, 1:400 for IF) Rabbit anti-β-catenin (CST, 8480; 1:400 for IF), mouse anti-α-catenin (13-9700; 1:400 for IF) antibodies was from Thermo Fisher Scientific. Secondary goat antibodies conjugated to Alexa Fluor 488, 568, or 647 (Thermo Fisher) were used at 1:1000 dilutions.

For visualization and quantitative analyses, monolayer MDCK cells were fixed with PBS containing 3.7% paraformaldehyde for 10 min followed by permeabilization with 0.2% Triton X-100 for 8 min. After blocking with PBS with 0.05% Tween-20 buffer containing 1% bovine serum albumin for 1 h at room temperature (RT), the fixed cells were incubated with primary antibodies in a humidified chamber for 1 h at RT or overnight at 4 °C, followed by secondary antibodies for 1 h at RT. DNA was stained with 4′,6-diamidino-2-phenylindole (DAPI) (Sigma). All images were collected using a Zeiss laser-scanning confocal microscope (Zeiss LSM 880). To generate side views of 2D monolayers, fixed and stained samples were scanned at a 0.5 μm interval at the Z axis from the apical to basal membrane and reconstructed by an LSM 880 microscope. The laser wavelengths used were 405 nm, 488 nm, 561 nm, and 647 nm, and the laser intensity was kept to a minimum. Image analysis and fluorescence intensity measurements were performed with ImageJ (https://imagej.net/software/imagej/). Bar graph figures were generated using Graphpad Prism (version 8.0.1). The experiments were repeated at least three times.

## Western blotting

For western blotting assays, MDCK cells depleted of endogenous E-cadherin were harvested and lysed in RIPA buffer (20 mM Tris-HCl, 150 mM NaCl, 1% Triton X-100, 0.5% sodium deoxycholate, 0.1% SDS, and 2 mM EDTA at pH 7.5). Samples were heated at 95 °C for 5 min, separated by SDS–PAGE, and analyzed by immunoblotting on polyvinylidene fluoride membranes. Rabbit polyclonal antibodies against E-cadherin (1:1000, Proteintech 20874-1-AP) were used. HPR-conjugated goat anti-rabbit secondary antibodies were used, followed by exposure to autoradiography film. The experiments were repeated at least three times.

## Protein isotope labeling for NMR studies

Isotopically labeled protein samples were expressed in corresponding minimal $M9/H_2O$ or $M9/D_2O$ medium. Uniformly $^{15}N$-labeled (or $^{13}C$- or $^{15}N$-labeled) protein samples were prepared in $M9/H_2O$ medium supplemented with 1 g liter$^{-1}$ of $^{15}NH_4Cl$ and 2 g/L of glucose (or $^{13}C$-glucose). $^{2}H$-Methyl-$^{13}CH_3$-Met-labeled samples were prepared according to published methods[35,36]. The cells were harvested at $OD_{600}$ ~ 1.0. The precursors or isotope-labeled amino acids used in this study were purchased from Cambridge Isotope Laboratories (CIL).

## NMR experiments

All NMR samples were prepared in buffer (100 mM phosphate, 100 mM NaCl at pH 7.2). HSQC measurements were performed on a Bruker Avance 600 spectrometer equipped with a cryo-probe. The concentrations of protein samples were ~100 μM, and data were acquired at 298 K (for methyl-trosy experiments) or 283 K (for E-cadherin CTD). Backbone resonance assignments of proerin were achieved by using standard triple-resonance experiments. All NMR spectra were processed using the NMRPipe program and analyzed using NMRViewJ software (https://nmrfx.org/nmrfx/nmrviewj).

## Protein crystallization and structure determination

Crystals of the E-cadherin–AnkR complex were obtained by the hanging drop vapor diffusion method at 16 °C under the conditions of 25% pentaerythritol ethoxylate (3/4 EO/OH), 0.1 M Tris pH 7.5. Glycerol (20%) was added as the cryoprotectant before diffraction data collection. The diffraction data were collected at the Shanghai Synchrotron Radiation Facility at 100 K and processed using HKL3000. The structure was solved by PHASER software [https://www.ccp4.ac.uk/html/phaser.html] using the molecular replacement method with the structure of AnkB R8-14 (Protein Data Bank: 5Y4E) as the search model. Further model modifications and refinements were repeated alternatively using COOT [https://www2.mrc-lmb.cam.ac.uk/personal/pemsley/coot/] and PHENIX [http://www.phenix-online.org/]. The final model was validated using MolProbity, and the statistics are shown in Table S1. The structure figures were made using PyMol [https://pymol.org/2/].

## Reporting summary

Further information on research design is available in the Nature Portfolio Reporting Summary linked to this article.

# Data availability

All data needed to evaluate the conclusions in the paper are present in the paper and/or the Supplementary Materials. Data supporting the findings of this manuscript are available from the corresponding author upon request. A reporting summary for this article is available as a Supplementary Information file. The source data underlying Figs. 6i, 6j, 6l, Figs. 7s, 7t, 7u and Supplementary Figs. 7i and 8a are provided as a Source Data file. Source data are provided with this paper.

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

## Acknowledgements

We thank Drs. Mingjie Zhang, Xiaotian Liu and Zhiyi Wei (Southern University of Science and Technology) for helpful discussion, Drs. Xuebiao Yao, Xing Liu and Xueying Wang (University of Science and Technology of China) for the assistance in cellular assays, and members of the Shanghai Synchrotron Radiation Facility (SSRF, China; beamlines BL02U1, BL18U1, and BL19U1) for X-ray beam time. This work was supported by grants from the National Natural Science Foundation of China (32170767, 22122703, 32100764, 91953110, and 31670734, to C.W.; 31971144, T2221005, and 31770807, to C.H.), the Innovative Program of the Development Foundation of Hefei Center for Physical Science and Technology (2018CXFX008, to C.W.), the Center for Advanced Interdisciplinary Science and Biomedicine of IHM (QYPY20220014, to C.W.), the Ministry of Science and Technology of the People's Republic of China (2019YFA0508402, to C.W.), the Strategic Priority Research Program of the Chinese Academy of Sciences (XDB0490000, to C.W.), USTC Research Funds of Double First-Class Initiative (YD9100002006, to C.W.), and the Fundamental Research Funds for the Central Universities (WK9100000029, WK9100000013, to C.W.).

## Author contributions

C.W., C.H., J.L., and C.K. designed the research; C.K., X.Q., D.C., W.X., Y.Z., S.Z., and M.L. performed biochemical experiments; C.K. and J.L. carried out the X-ray data collection; C.K. performed the cellular experiments with help from Z.L. and F.Z.; C.H. and X.Q. performed the NMR experiments; J.L. determined the crystal structure; all authors analyzed the data; C.W., C.H., and C.K. drafted the manuscript, and all authors approved the manuscript; C.W. coordinated the project.

## Competing interests

The authors declare no competing interests.
