## [Peer Review File · Nature Communications]

Dynamic interactions between E-cadherin and Ankyrin-G mediate epithelial cell polarity maintenanceREVIEWER COMMENTS

Reviewer #1 (Remarks to the Author):

This study presents a comprehensive structural and functional analysis of interaction of ankyrin-G (ankG) with the cytoplasmic domain of e-cadherin (ECD). Three ankG binding-sites were identified in the unstructured ECD using NMR with N15-labeled ECD, where site one overlaps with the site for p120 catenin, and sites 2 and three overlap with the beta-catenin site (previously mapped by Huber and Weiss). Evidence for a common docking site for all 3 ankG-binding motifs of ECD was provided by Methyl-TROSY NMR using ankG bearing C13-labeled methionine residues combined with NMR analysis of an ankG-ECD fusion construct. Critical aromatic residues in each site were identified that are required for binding of ankG as well as p120 catenin, and beta-catenin. Interestingly, while p120 catenin and beta catenin each required either site one (p120 catenin) or sites two/three (beta-catenin) for full activity, ankG required all three sites. Expression of e-cadherin constructs bearing mutated ECD aromatic residues in MDCK cells revealed that that mutation of residues in all three sites is required to eliminate accumulation of e-cadherin and ankG at the lateral membrane. The authors interpret these and other experiments as evidence for a dynamic interaction of all three ECD sites with a shared pocket on ankG.

The concept of a dynamic binding pocket that can encompass diverse unstructured peptides is original and potentially of broad significance. Such "fuzzy" interactions would not be resolved by current structural methods, and may turn out to be prevalent. I have a few suggestions/concerns that the authors should address:

1. Given that the affinity for beta-catenin binding to ECD is about 1000 times stronger than binding of either ankG or p120, a beta-catenin-ECD complex is the likely physiological entity. If this is the case, does ankG ever encounter sites two/three on the ECD in the context of a cell?
2. Gel filtration measures the Stokes radius and not the molecular weight of polymers. Molecular weights can be deduced from elution profiles only if the sedimentation coefficient is known or if the polymers are known to be configured as tightly packed spheres. The ECD is a random coil, and ankG is an extended solenoid. The elution profiles thus are difficult to interpret, and estimates of stoichiometry have serious caveats.
3. Figure 5 evaluates functional consequences of aromatic residue mutations in sites one, two, or three (and various combinations of sites). These mutations would affect ankG binding as well as interactions with p120 and beta catenin (shown in fig 6). It would be easier to follow the logic if fig 6 came first, and there was a schematic summarizing effects of the mutations on all three proteins.

Reviewer #2 (Remarks to the Author):

This manuscript from Kong and colleagues describes their efforts to characterize the interaction between E-cadherin and Ankyrin-G. E-cadherin has been postulated to form a complex with Ankyrin-G to regulate epithelial cell polarity. The molecular details of this interaction and the importance of this complex in epithelial polarity are poorly defined. Here, the authors apply a number of biochemical approaches (SEC-MALS, NMR, ITC) using purified proteins to map the binding interface and measure the affinity between the E-cadherin cytosolic tail and Ankyrin-G. They find that Ankyrin-G can bind to three separate regions of the E-cadherin tail to form "dynamic" complexes that can exceed 1:1 stoichiometry. They identify key aromatic residues within each E-cadherin binding site that, when mutated, limit or block Ankyrin-G binding. They also show that E-cadherin, beta-catenin and Ankyrin-G can form a complex in solution. Finally, they show that in cells mutations in the E-cadherin tail can reduce Ankyrin-G recruitment and disrupt polarization in MDCK epithelial cells

Overall, this is a comprehensive and largely convincing analysis of the E-cadherin-Ankyrin interaction. The manuscript is logical, well-written and clear. A tremendous amount of work went into this story and the authors are to be commended for their Herculean efforts. The ITC experiments alone (well over 30 different protein-ligand combinations!) are a true tour-de-force. I have some minor quibbles and a couple of questions about the data, detailed below. My biggest concerns, however, involve the depth of the study, the relevance of the interaction, and the larger impact of the work. The story, for

all the work that went into it, maps the E-cadherin-Ankyrin G interface in great deal but does little else. There are interesting facets to this "dynamic interaction" as the authors refer to it, but I'm not convinced by the novelty of the interface. Also, the biological relevance/importance of the "dynamic interaction" is largely untested. It is nice study that should interest though who work on Ankyrin-G and perhaps E-cadherin, but I don't think it will be appeal to a wider scientific audience.

Concerns (in no particular order):

- The relative weak affinity of the E-cadherin tail for Ankyrin-G (~ 1 micromolar) compared to the strong affinity for b-catenin (~ 30 nanomolar) leads me to question just how physiologically relevant this interaction is. "Dynamic" is one way to frame the weak interaction, but the model proposed in Fig 6E where Ankyrin-G binds all cadherin-catenin complexes seems unlikely unless the concentration of Ankyrin-G is in excess of 10 micromolar at the membrane. Perhaps it is that high in certain regions? Or perhaps other binding partners (spectrin?) or post-translational modifications increase the affinity of Ankyrin-G for E-cadherin?
- The amino acid numbering of E-cadherin 1A is incorrect. The E-cadherin propeptide (before N-terminal cleavage) is 884 amino acids. The mature peptide (which starts at EC1 as depicted in the cartoon) is 728 amino acids. Please double check your amino acid designations throughout the manuscript.
- The "n" for all ITC experiments should be listed in the isotherm and/or summary table. Given the large number of runs performed, I assume that some may have only been done once or twice, but the n should be provided for all.
- In Fig 6A, the kDa values don't match the right y-axis (the presence of which is confusing honestly). Also, why is E-cad/b-catenin complex a lower MW than b-catenin alone?
- The presence of multiple Ankyrin-G binding sites in the E-cadherin tail raises the possibility of cooperative binding. Did the authors detect any evidence of cooperative binding in the ITC (or other) experiments?
- In Fig 5, the phenotypes observed with the E-cadherin tail mutants could all be caused by loss of cadherin-mediated adhesion (via disruption in p120 and/or b-catenin recruitment) rather than loss of Ankyrin-G recruitment, as suggested. The authors should measure and compare beta-catenin and p120-catenin (and ideally alpha-E-catenin) recruitment in A-H and K.
- I think it might be informative to analyze the dynamics of the E-cadherin tail mutants by FRAP at cell-cell contacts in MDCK cells, though it could prove difficult to separate differences based on Ankyrin-G recruitment versus catenin recruitment.

Reviewer #3 (Remarks to the Author):

The manuscript, Dynamic interactions between E-cadherin and Ankyrin-G, mediate epithelial cell polarity maintenance by Kong et al., describes the evaluation of binding between the two proteins using bioanalytical, structural biology, and cellular assays. Authors have used several experimental methods to prove that E-cadherin and Ankyrin-G interact with three different sites on E-cadherin. The interaction of these two proteins was proposed more than a decade ago. However, the dynamic binding nature of these proteins and proof of three distinct binding sites on E Cadherin is something new. The manuscript is acceptable for publication in Nature Communications after the authors address the following comments.

1. Most of the experimental evidence to prove the interaction of the two proteins (E-cadherin and Ankyrin-G) is performed using purified proteins and fragments of polypeptides using ITC and NMR experiments. The evidence of E-cadherin variants binding to Ank-G or in cells that are deficient is only for accumulation at the lateral membrane. The reviewer is aware that details of structural evidence are obtained using structural biology methods. Did the authors try proximity ligation assay to show

that mutant constructs of E-Cadherin bind/do not bind to verify the protein-protein interactions in the cellular environment?

2. E-cadherin trafficking is known to occur in the membrane vesicles. How does the trafficking effect the dynamic nature of the binding of E-cadherin and Ankrin-G ? The authors should briefly discuss this in light of their new findings.

3. In the discussion section (lines 416-429), the authors discuss the details of the binding site, the amino acids involved, and the hydrophobic nature of the binding sites. Protein-Protein interaction sites are dominated by hydrophobic amino acids. Authors can compare the PPI sites and amino acids in PPI sites reported in the literature and compare the dynamic and specific binding sites reported in their study regarding amino acids in these sites.

4. The authors mention that the crystallographic data quality was poor and have not discussed the details of the binding site using the X-ray data. Was this due to the dynamic nature of the binding site, or the protein constructs did not acquire a proper secondary structure because of the expression of particular domains of the disordered protein?

Reviewer #4 (Remarks to the Author):

Dynamic interactions between E-cadherin and Ankyrin-G mediate epithelial cell 2 polarity maintenance

Kong...Wang, Nat. Comm., 2023

Epithelial cadherin (E-cadherin) is a cell-cell adhesion protein that plays an essential role in the formation of adherens junctions, and therefore in maintaining the stability and organization of epithelial lateral membranes and the maintenance of cell polarity. E-cadherin is tethered to the cytoskeleton, via its C-terminal domain, to β -catenin, forming the canonical cadherin-catenin complex to connect with the actin-based cytoskeleton. E-cadherin also directly binds to Ankyrin-G (AnkG), a scaffold protein that links membrane proteins to the spectrin-based cytoskeleton. However, the mechanism by which this complex is formed and its relationship with the cadherin-catenin complex is unknown.

Kong et al use isothermal titration calorimetry, analytical gel filtration chromatography and NMR to determine that E-cadherin binds to AnkG with a stoichiometry ranging from 1:1 to 1:2. They identify three distinct sites on E-cadherin that can bind to the same binding region on AnkG, and show that binding by the three E-cadherin sites to AnkG is dynamic in nature, with all three E-cadherin sites competing to occupy the single pocket on AnkG. The authors also show that disruption of all three bonding sites E-cadherin is necessary to abrogate binding to AnkG and furthermore, that this leads to disruption of cell polarity. Finally, the authors show that E-cadherin can bind to both AnkG and β -catenin simultaneously, providing mechanistic insights into the formation of the ankyrin-spectrin complex alongside the cadherin-catenin complex.

The study describes a previously unknown phenomenon – the promiscuous and dynamic nature of association between E-cadherin and AnkG, and the authors suggest that disrupting this phenomenon might affect cellular polarity.

However, the study so far raises several intriguing questions that have not been gone into in sufficient detail. The manuscript could be improved substantially by additional discussion/analysis/figures to flesh out few points, discussed below.

- Both E-cadherin and AnkG are associated with and function at the membrane, yet the study does not explore whether the dynamic nature of these association between these two proteins differs at a membrane. The authors show that disruption of all three binding sites on E-cadherin is necessary for disruption of cell polarity. However, this might not necessarily be (or only be) because of the disruption of the association between E-cadherin and AnkG. Interactions between the membrane and one or both proteins might also play a role. It would be good if the authors could comment on this.

- The authors have not explored the functional implications of the dynamic nature of the interaction between E-cadherin and AnkG or discussed why the dynamic nature of this interaction necessary for

the maintenance of cellular polarity. It would be good to at least include a description of these implications in the Discussion section.

- A few minor points:

- o In the section "Discrete sites within E-cadherin bind to AnkG in a dynamic manner", it would be good if the authors could include a description of how the various truncation fragments were determined.

- o In the section "Promiscuous engagement of E-cadherin by AnkG" the authors discuss ITC analyses of three truncations (R6-24, R8-24 and R8-14) of AnkG with the entire C-terminal domain of E-cadherin (CTD) and also with E-cadherin site 1 (residues 751-775) and E-cadherin site 2,3 (residues 828-869). However there doesn't appear to be a figure/table showing the ITC data of R8-24 with the E-cadherin CTD. It would be good to include this.

- o In Figure 5a-h, it would be good to include the data from the cells expressing shRNA targeting E-cadherin, but which have not been transfected with any GFP-E-cadherin, both as a control and to verify that AnkG does not accumulate at the lateral membrane in the absence of E-cadherin.

Point-by-point responses to the comments raised by the reviewers

Before point-by-point responses to the referees' comments, we thank all the reviewers for recognizing the novelty and interests of our works and their critical and constructive suggestions and guidance that help us to efficiently improve our manuscript. Our responses are shown in blue.

Reviewer #1 (Remarks to the Author):

This study presents a comprehensive structural and functional analysis of interaction of ankyrin-G (ankG) with the cytoplasmic domain of e-cadherin (ECD). Three ankG binding-sites were identified in the unstructured ECD using NMR with N¹⁵-labeled ECD, where site one overlaps with the site for p120 catenin, and sites 2 and three overlap with the beta-catenin site (previously mapped by Huber and Weiss). Evidence for a common docking site for all 3 ankG-binding motifs of ECD was provided by Methyl-TROSY NMR using ankG bearing C¹³-labeled methionine residues combined with NMR analysis of an ankG-ECD fusion construct. Critical aromatic residues in each site were identified that are required for binding of ankG as well as p120 catenin, and beta-catenin. Interestingly, while p120 catenin and beta catenin each required either site one (p120 catenin) or sites two/three (beta-catenin) for full activity, ankG required all three sites. Expression of e-cadherin constructs bearing mutated ECD aromatic residues in MDCK cells revealed that that mutation of residues in all three sites is required to eliminate accumulation of e-cadherin and ankG at the lateral membrane. The authors interpret these and other experiments as evidence for a dynamic interaction of all three ECD sites with a shared pocket on ankG.

The concept of a dynamic binding pocket that can encompass diverse unstructured peptides is original and potentially of broad significance. Such “fuzzy” interactions would not be resolved by current structural methods, and may turn out to be prevalent. I have a few suggestions/concerns that the authors should address:

We would like to express our gratitude to the reviewer for the comprehensive summary and high praise of our work. As highlighted by the reviewer, the dynamic interaction between E-cadherin and AnkG, along with the utilization of NMR approaches, are the key aspects we aim to emphasize. We have taken great care to address all the suggestions and concerns raised by the reviewer in a thorough manner.

1. Given that the affinity for beta-catenin binding to ECD is about 1000 times stronger than binding of either ankG or p120, a beta-catenin-ECD complex is the likely physiological entity. If this is the case, does ankG ever encounter sites two/three on the ECD in the context of a cell?

We appreciate the reviewer for posing this question. The binding between beta-catenin and ECD has been extensively studied for over a decade using well-designed structural and functional experiments. Notably, the binding affinity of beta-catenin with ECD is significantly stronger compared to both AnkG and p120 catenin. Moreover, the region on ECD responsible for beta-catenin binding is relatively long, encompassing both sites 2 and 3, spanning approximately 100 amino acids. Nevertheless, we believe that AnkG can still interact with sites 2/3 on the ECD both *in*

in vitro and under cellular conditions, as supported by our ITC experiments and cell-based studies. Firstly, we confirmed the formation of a tertiary complex through ITC experiments using titrations of E-cadherin 776-884 (sites 2/3) and AnkG R1-24 complex with beta-catenin (Figure 1a). The interaction remained intact when E-cadherin 776-884 and beta-catenin complex were titrated with AnkG R1-24 (Figure 1b). Since site-1 (751-775) on ECD was not included, the formation of the three-subunit complex demonstrates that AnkG can interact with E-cadherin on sites 2/3 in the presence of beta-catenin. Secondly, in our knockdown and rescue experiments, we observed that upon transfection of GFP-E-cadherin site-1 to A variant into MDCK cells to restore E-cadherin expression, the variant still exhibited membrane localization with minimal mis-sorting to the cytosol, and AnkG was also restricted to the membrane region (Figure 1c-f). These results indicate that AnkG can bind to the E-cadherin site-1 to A variant, suggesting that AnkG interacts with sites 2/3 within a cellular context.

Figure 1. (a) ITC result of E-cadherin 776-884 (sites 2/3) and AnkG R1-24 complex to β -catenin. (b) ITC result of E-cadherin 776-884 and β -catenin complex to AnkG R1-24. (c-d) MDCK cells stably expressing a shRNA targeting E-cadherin were transfected with wild-type (WT) GFP-E-cadherin (c), and GFP-E-cadherin site-1 to A (d). The cells were stained with GFP signals showing E-cadherin localization, endogenous AnkG signals showing AnkG localization, and actin signals showing the lateral membranes. Scale bars: 10 μm . (e-f) Quantification of the immunofluorescence

intensity ratios of cytosolic to membranes for E-cadherin WT (n=70); site-1 to A (n=45); site-2 to A (n=48); site-3 to A (n=52); site-1, 2 to A (n=39); site-1, 3 to A (n=41); site-2, 3 to A (n=42); and site-1, 2, 3 to A (n=35) (e) ; or AnkG in the groups of WT (n=41); site-1 to A (n=40); site-2 to A (n=37); site-3 to A (n=44); site-1, 2 to A (n=43); site-1, 3 to A (n=53); site-2, 3 to A (n=55); site-1, 2, 3 to A (n=43); and GFP control (n=25) (f). Data are presented as the means \pm SEMs and analyzed using one-way ANOVA followed by Dunnett's multiple comparisons test to WT E-cadherin or AnkG , *p<0.05, ***p<0.001, ****p<0.0001, ns, not significant.

2. Gel filtration measures the stokes radius and not the molecular weight of polymers. Molecular weights can be deduced from elution profiles only if the sedimentation coefficient is known or if the polymers are known to be configured as tightly packed spheres. The ECD is a random coil, and ankG is an extended solenoid. The elution profiles thus are difficult to interpret, and estimates of stoichiometry have serious caveats.

We fully appreciate the guidance provided by the reviewer regarding the elution profiles of the gel filtration assay, which measures the Stokes radius rather than the direct molecular weights. That is precisely why we employed a combination of size-exclusion chromatography (SEC) and multi-angle light scattering (MALS) assays in our studies to investigate molecular weights and to estimate stoichiometry. This approach offers a robust technique for the absolute characterization of macromolecular weights by incorporating a multi-angle light scattering (MALS) photometer. In SEC-MALS, the SEC column functions to separate molecules based on their hydrodynamic volume, with the retention time not being used to determine molecular weight. Unlike conventional SEC, where the molar mass is always relative to calibration standards, the molar mass values obtained through MALS are absolute, relying on the theoretical first principles relationship between molar mass, intensity of scattered light, and concentration. Therefore, we firmly believe that our results from SEC-MALS analysis are valid given the circumstances.

3. Figure 5 evaluates functional consequences of aromatic residue mutations in sites one, two, or three (and various combinations of sites). These mutations would affect ankG binding as well as interactions with p120 and beta catenin (shown in fig 6). It would be easier to follow the logic if fig 6 came first, and there was a schematic summarizing effects of the mutations on all three proteins.

We express our gratitude to the reviewer for providing these valuable suggestions. Following the reviewer's guidance, we have now revised our manuscript. Firstly, we have reorganized the presentation of our data by moving the original Figure 6c and 6d into a new table (Table I). This allows us to introduce these results prior to the cellular experiments, enhancing the overall flow of the manuscript. Furthermore, we have included additional information regarding the ITC experiments and provided a comprehensive summary of the effects of these E-cadherin mutations on binding with all three proteins: AnkG, beta-catenin, and p120 catenin.

a				b			
E-cadherin 734-884 (CTD)	β -catenin 1-781 K_d (μ M)	n value	fold change	E-cadherin 734-884 (CTD)	p120 catenin 324-938 K_d (μ M)	n value	fold change
WT	0.036 ± 0.004	0.80	1	WT	1.7 ± 0.1	0.92	1
site-1 to A	0.056 ± 0.004	0.98	~ 1.5	site-1 to A	>50	-	>30
site-2 to A	4.8 ± 0.6	0.76	>100	site-2 to A	2.8 ± 0.1	0.5	~ 2
site-3 to A	0.11 ± 0.01	0.85	~ 3.1	site-3 to A	2.9 ± 0.1	0.84	~ 2
site-1,2 to A	3.6 ± 0.3	0.77	>100	site-1,2 to A	>70	-	>40
site-1,3 to A	0.098 ± 0.004	0.72	~ 2.7	site-1,3 to A	>30	-	~ 20
site-2,3 to A	1.4 ± 0.2	0.81	~ 28	site-2,3 to A	1.8 ± 0.1	0.8	~ 1
site-1,2,3 to A	8.1 ± 0.7	0.74	>200	site-1,2,3 to A	>50	-	>30

Table I. (a) ITC binding assays showing the measured binding affinities between β -catenin 1-781 and the indicated E-cadherin CTD WT and variants. The n values and the fold changes of K_d compared to the E-cadherin CTD WT are shown. (b) ITC binding assays showing the measured binding affinities between p120 catenin 324-938 and the indicated E-cadherin CTD WT and variants. The n values and the fold changes of K_d compared to the E-cadherin CTD WT are shown.

Reviewer #2 (Remarks to the Author):

This manuscript from Kong and colleagues describes their efforts to characterize the interaction between E-cadherin and Ankyrin-G. E-cadherin has been postulated to form a complex with Ankyrin-G to regulate epithelial cell polarity. The molecular details of this interaction and the importance of this complex in epithelial polarity are poorly defined. Here, the authors apply a number of biochemical approaches (SEC-MALS, NMR, ITC) using purified proteins to map the binding interface and measure the affinity between the E-cadherin cytosolic tail and Ankyrin-G. They find that Ankyrin-G can bind to three separate regions of the E-cadherin tail to form "dynamic" complexes that can exceed 1:1 stoichiometry. They identify key aromatic residues within each E-cadherin binding site that, when mutated, limit or block Ankyrin-G binding. They also show that E-cadherin, beta-catenin and Ankyrin-G can form a complex in solution. Finally, they show that in cells mutations in the E-cadherin tail can reduce Ankyrin-G recruitment and disrupt polarization in MDCK epithelial cells.

Overall, this is a comprehensive and largely convincing analysis of the E-cadherin-Ankyrin interaction. The manuscript is logical, well-written and clear. A tremendous amount of work went into this story and the authors are to be commended for their Herculean efforts. The ITC experiments alone (well over 30 different protein-ligand combinations!) are a true tour-de-force. I have some minor quibbles and a couple of questions about the data, detailed below. My biggest concerns, however, involve the depth of the study, the relevance of the interaction, and the larger impact of the work. The story, for all the work that went into it, maps the E-cadherin-Ankyrin G interface in great deal but does little else. There are interesting facets to this "dynamic interaction" as the authors refer to it, but I'm not convinced by the novelty of the interface. Also, the biological relevance/importance of the "dynamic interaction" is largely untested. It is nice study that should interest though who work on Ankyrin-G and perhaps E-cadherin, but I don't think it will be appeal to a wider scientific audience.

We extend our gratitude to the reviewer for his comments and for recognizing the extensive efforts we have invested in unraveling the molecular basis of the E-cadherin-Ankyrin-G interaction. As acknowledged by the reviewer, we engaged in thorough brainstorming to establish the correlation between the dynamic nature of complex formation and its functional consequences within cellular contexts. Our perspective is that the maintenance of the E-cadherin complex at adhesion junctions is not solely reliant on the previously characterized cadherin-catenin complex; ankyrins likely play a crucial role in stabilizing E-cadherin on lateral membranes. Through our present study, which delves into the interface between E-cadherin and AnkG, we propose that the dynamic interaction mechanism ensures the association between E-cadherin and AnkG. Moreover, we believe that the promiscuous binding mode itself introduces distinct features and novelties to both the AnkG and E-cadherin fields. Once again, we express our appreciation to the reviewer for their valuable guidance in enhancing our manuscript.

Concerns (in no particular order):

- The relative weak affinity of the E-cadherin tail for Ankyrin-G (~1 micromolar) compared to the

strong affinity for β -catenin (~30 nanomolar) leads me to question just how physiologically relevant this interaction is. "Dynamic" is one way to frame the weak interaction, but the model proposed in Fig 6E where Ankyrin-G binds all cadherin-catenin complexes seems unlikely unless the concentration of Ankyrin-G is in excess of 10 micromolar at the membrane. Perhaps it is that high in certain regions? Or perhaps other binding partners (spectrin?) or post-translational modifications increase the affinity of Ankyrin-G for E-cadherin?

We thank the reviewer for this thought-provoking question. Investigating the interactions among proteins always revolves around their physiological relevance, which is of utmost importance. The discovery of the dynamic mode presented here is surprising, as it is novel for both the ankyrin-mediated membrane complex and the canonical cadherin-catenin complex. While the binding affinity between E-cadherin and AnkG is relatively weaker compared to cadherin-catenin interactions, we believe it holds significance under physiological conditions, and the observed ~1 micromolar Kd *in vitro* does not necessarily indicate a weak interaction. This is supported by several previous studies demonstrating the essential functional roles of the E-cadherin-Ankyrin-G complex in cell cultures and mouse models^[1]. In Figure 6e, our proposed model aims to convey the message that AnkG can consistently be tethered to the E-cadherin tail through the dynamic nature of the binding. Additionally, inspired by the reviewer's insights, we also speculate that the concentration of Ankyrin-G, as a central scaffold protein linking membrane proteins to the spectrin-based cytoskeleton, may be sufficiently high to enable effective binding to E-cadherin at the membranes. Taking the synaptic scaffold protein SynGAP as an example^[2]; in addition to its activity as a Ras/Rap GAP, SynGAP (~135 kDa) contains several motifs for protein-protein interaction. Cheng et al. employed an absolute quantification strategy using synthetic isotope-labeled peptides as internal standards to measure the molar abundance of 32 key PSD proteins in the forebrain and cerebellum. They found that SynGAP (~2.1 pmol/20 μ g) was approximately as abundant as the scaffold proteins it interacts with, such as the PSD-95 family (which totaled ~2.3 pmol/20 μ g)^[3] (measurements were made in ~20 μ g of forebrain or cerebellar PSD sample). This indicates the abundance of scaffold proteins in organizing specific micro membrane domains. Considering the extremely small volume of the PSD, the concentrations of SynGAP, as well as PSD95, are in the range of several micromolar tiers. Although it has not been determined yet, AnkG, being the master scaffold protein linking diverse membrane proteins to the cytoskeleton, is believed to be highly abundant beneath the membranes. While phosphorylation has been reported to regulate the complex formations of E-cadherin- β -catenin and Ankyrin-G-Neurofascin, whether other binding partners (e.g., spectrin) or post-translational modifications could modulate the affinity of AnkG for E-cadherin has yet to be investigated.

- The amino acid numbering of E-cadherin 1A is incorrect. The E-cadherin propeptide (before N-terminal cleavage) is 884 amino acids. The mature peptide (which starts at EC1 as depicted in the cartoon) is 728 amino acids. Please double check your amino acid designations throughout the manuscript.

We thank the reviewer for pointing out our mistakes and we sincerely apologize for the oversight in the amino acid numbering of E-cadherin. As guided by the reviewer, we have rectified our E-cadherin domain organization in the cartoon (Figure II). To maintain consistency throughout the

manuscript, we have adopted the propeptide numbering (884 amino acids) in all aspects of our study.

Figure II. Schematic diagram showing the domain organizations of E-cadherin and AnkG. EC, extracellular cadherin domains; TM, transmembrane domain; CTD, C-terminal domain; ANK repeats, Ankyrin repeats; DD, death domain.

- The "n" for all ITC experiments should be listed in the isotherm and/or summary table. Given the large number of runs performed, I assume that some may have only been done once or twice, but the n should be provided for all.

Thank the reviewer for the valuable suggestions. We have now incorporated "n" values for all the ITC experiments in the revised manuscript. The ITC technique is very robust for experienced researchers. We want to emphasize that we have diligently repeated the pivotal ITC experiments multiple times, ensuring the reliability and accuracy of the results presented in our study.

- In Fig 6A, the kDa values don't match the right y-axis (the presence of which is confusing honestly). Also, why is E-cad/b-catenin complex a lower MW than b-catenin alone?

In order to enhance clarity, we have made modifications to Fig 6a, presenting a more concise illustration of the three-subunit complex formation. Regarding the molecular weights fitting, we too were somewhat surprised that the E-cad/b-catenin complex appeared lower than b-catenin alone. However, the gel filtration profile provided clear evidence of complex formation. While the selection of the peak area is a variable factor in calculating the MW, its impact is minimal. To ensure accuracy, we conducted additional experiments by mixing the E-cadherin and b-catenin in ratios of 1:1 and 1.5:1, respectively. In the case of the 1:1 mixture, the measured molecular weight of the complex was ~94.0 kDa (Figure IIIa), higher than b-catenin alone, but still lower than the expected MW. In the 1.5:1 mixture, the measured molecular weight of the complexes was ~101.3 kDa (Figure IIIb), indicating a greater tendency toward complex formation. Nevertheless, it is important to emphasize that the SEC-MALS is still a powerful technique for accurately measuring the molecular weights of protein complexes.

Figure III. (a-b) Analytical gel filtration chromatography analysis coupled with static light scattering analysis of E-cadherin CTD (orange), β -catenin 1-781 (cyan) and E-cadherin CTD- β -catenin 1-781 complex (magenta) with the ratios of E-cadherin: β -catenin at 1:1 (a) or 1.5:1 (b).

- The presence of multiple Ankyrin-G binding sites in the E-cadherin tail raises the possibility of cooperative binding. Did the authors detect any evidence of cooperative binding in the ITC (or other) experiments?

Thank the reviewer for the question. Initially, we also considered the possibility of cooperative binding due to the presence of multiple AnkG binding sites in the E-cadherin tail. However, our experimental findings did not indicate the cooperative binding activities as the binding affinity for the entire E-cadherin CTD encompassing all three sites, or the E-cadherin 776-834 fragment containing sites 2 and 3 with AnkG was comparable to that of the individual binding sites within the E-cadherin CTD.

- In Fig 5, the phenotypes observed with the E-cadherin tail mutants could all be caused by loss of cadherin-mediated adhesion (via disruption in p120 and/or β -catenin recruitment) rather than loss of Ankyrin-G recruitment, as suggested. The authors should measure and compare β -catenin and p120-catenin (and ideally α -E-catenin) recruitment in A-H and K.

We thank the reviewer for these helpful suggestions. In this study, we established a stable cell line expressing a shRNA targeting on E-cadherin. Surprisingly, we observed that the loss of E-cadherin had no discernible effects on the localization of β -catenin or p120 catenin. This was evident from staining of the endogenous β -catenin (Figure IVa) or the endogenous p120 catenin (Figure IVb) in MDCK cells. These findings strongly suggested that the phenotypic changes observed in our experiments with the E-cadherin tail mutants were primarily a consequence of the disrupted binding between E-cadherin and AnkG, which subsequently led to the impairment of polarity maintenance.

Figure IV. Loss of E-cadherin had no effect on β -catenin or p120 catenin localization. MDCK cells stably expressing an E-cadherin-targeting shRNA and stained with endogenous β -catenin (a) or stained with endogenous p120-catenin (b).

Consistent with our cellular results, previous reports have also indicated that depletion of E-cadherin by virus-transfected shRNA in MDCK cells did not disrupt the localization of β -catenin and α -catenin at the adherens junctions where cell–cell contacts (Figure V). However, it is worth noting that these cells did undergo noticeable morphological changes^[4].

Figure V. Cell–cell contacts are maintained in cells depleted of E-cadherin. Immunofluorescence staining of MDCK cells fixed 2 d post-transfection with E-cadherin pS Ecad or pS Luc shRNAs. YFP expression is a transfection marker. Adherens junction proteins E-cadherin, β -catenin, α -catenin, and actin localization at cell–cell contacts in both control and knockdown cells^[4].

- I think it might be informative to analyze the dynamics of the E-cadherin tail mutants by FRAP at cell-cell contacts in MDCK cells, though it could prove difficult to separate differences based on Ankyrin-G recruitment versus catenin recruitment.

We agree with the reviewer’s suggestions regarding the analysis of E-cadherin variants through FRAP in MDCK cells, which could provide more information regarding E-cadherin complex formation. We acknowledge that distinguishing of the specific origins of the observed differences may pose a challenge. Additionally, conducting these experiments may also present technical

difficulties, as precise control of the expression levels of the transfected E-cadherin tail variants is necessary, and the areas for photobleaching of the membrane-anchored E-cadherin are relatively small. Nonetheless, we hope that the reviewer will agree with us that our MDCK cell data effectively demonstrates the essential roles of the E-cadherin-AnkG complex in the organization of lateral membranes.

Reviewer #3 (Remarks to the Author):

The manuscript, Dynamic interactions between E-cadherin and Ankyrin-G, mediate epithelial cell polarity maintenance by Kong et al., describes the evaluation of binding between the two proteins using bioanalytical, structural biology, and cellular assays. Authors have used several experimental methods to prove that E-cadherin and Ankyrin-G interact with three different sites on E-cadherin. The interaction of these two proteins was proposed more than a decade ago. However, the dynamic binding nature of these proteins and proof of three distinct binding sites on E Cadherin is something new. The manuscript is acceptable for publication in Nature Communications after the authors address the following comments.

We thank the reviewer for the careful reading of our study and for recognizing the significance of our work. We have now addressed all the reviewer's concerns, and have revised our manuscript according to the guidance of the reviewer.

1. Most of the experimental evidence to prove the interaction of the two proteins (E-cadherin and Ankyrin-G) is performed using purified proteins and fragments of polypeptides using ITC and NMR experiments. The evidence of E-cadherin variants binding to AnkG or in cells that are deficient is only for accumulation at the lateral membrane. The reviewer is aware that details of structural evidence are obtained using structural biology methods. Did the authors try proximity ligation assay to show that mutant constructs of E-Cadherin bind/do not bind to verify the protein-protein interactions in the cellular environment?

We thank the reviewer for the suggestion. We totally agree with the reviewer that proximity ligation assay (PLA) is a powerful technique for investigating protein-protein interactions within the cellular environment. However, in our manuscript focused on the E-cadherin-AnkG interaction, we did not employ PLA. Instead, we utilized polarized MDCK cell lines to observe the accumulation of the proteins and examine the lateral membrane morphology. We believe our biochemical and biophysical approaches have provided sufficient molecular insights into the interaction. Furthermore, we would like to highlight the contributions of prior pioneering studies conducted by the Bennett lab (Duke University). These studies have evaluated the binding of E-cadherin to ankyrins using cell-based recruitment assays, in which GFP-tagged proteins were recruited to the plasma membranes when co-expressed with full length E-cadherin (Figure VI)^[5]. Notably, AnkG-GFP was localized in the cytoplasm when expressed alone but exhibited confinement to the plasma membranes when co-expressed with E-cadherin (Figure VI)^[5]. We believe these studies have provided convincing evidences for the existence of the E-cadherin-AnkG interaction in the cellular environment.

Figure VI. HEK 293-based plasma membrane recruitment assay shows that E-cadherin (EC) recruits AnkG-GFP to the plasma membranes. Scale bars, 5 μm .^[5]

2. E-cadherin trafficking is known to occur in the membrane vesicles. How does the trafficking effect the dynamic nature of the binding of E-cadherin and Ankrin-G? The authors should briefly discuss this in light of their new findings.

We thank the reviewer for the suggestion and this is an interesting point to be included. Ankyrin is reported to regulate trafficking of several membrane proteins including Na^+/K^+ ATPase^[6], suggesting that Ankyrin functions not only on the maintenance of membrane protein localization, but also on the establishment of the membrane protein polarity. There is no direct evidence that AnkG regulates E-cadherin trafficking, but it is possible that AnkG participates in the process as the dynamic nature of the binding may ensure the complex formation. Previous studies have shown that AnkG regulates E-cadherin endocytosis at the lateral membranes^[1]. We have included these in the discussion part as directed by the reviewer.

3. In the discussion section (lines 416-429), the authors discuss the details of the binding site, the amino acids involved, and the hydrophobic nature of the binding sites. Protein-Protein interaction sites are dominated by hydrophobic amino acids. Authors can compare the PPI sites and amino acids in PPI sites reported in the literature and compare the dynamic and specific binding sites reported in their study regarding amino acids in these sites.

We thank the reviewer for these suggestions. In our previous studies, we reported the high-resolution crystal structures of the AnkG ankyrin repeat (ANK repeat) domain in complex with its binding site from the Nfasc cytoplasmic tail or with a fragment from Nav1.2^[7,8]. Our structural analysis revealed that ANK repeats interacted with essential Nfasc residues, including F1202, E1204, and Y1212 (Figure VII)^[8]. Moreover, the structure of Nav1.2/AnkB_repeats_R1-9 reveals Glu1112 of Nav1.2 anchors Nav channel to ankyrins (Figure VII)^[7]. The binding sites in Nfasc and Nav1.2 contain charged and hydrophobic amino acids while the high salt buffer condition disrupts the formation of these two complexes. We evaluated the binding affinities between E-cadherin CTD, or E-cadherin 828-884 and AnkG R1-24, respectively, through ITC assays in the buffer containing 500 mM NaCl. The high salt concentration did not affect the assembly of the complex, indicating hydrophobic interactions are dominant for the interaction between E-cadherin and AnkG, although charged residues are also present in the PPI sites on E-cadherin.

Figure VII. Amino acids in PPI sites from Nfasc, Nav1.2, and E-cadherin that interact with AnkG.

The critical amino acids in PPI sites for various Ankyrin-G binding peptides are labeled with yellow (charged amino acids) and red (hydrophobic amino acids).

4. The authors mention that the crystallographic data quality was poor and have not discussed the details of the binding site using the X-ray data. Was this due to the dynamic nature of the binding site, or the protein constructs did not acquire a proper secondary structure because of the expression of particular domains of the disordered protein?

Thank the reviewer for this question. We think the poor crystallographic data was due to the dynamic nature of the binding site. Despite extensive efforts to explore various combinations and conditions for crystal screening and data collection, the specific interaction sites of E-cadherin have not been successfully resolved. We have been focused on the structural and functional characterization of ankyrins for many years, and based on our previous studies of ankyrin proteins, we are confident that the protein constructs we used can acquire proper secondary structures necessary for their interactions. However, due to the dynamic nature of the E-cadherin-AnkG binding, obtaining high-resolution complex crystallographic data has been challenging.

Reviewer #4 (Remarks to the Author):

Epithelial cadherin (E-cadherin) is a cell–cell adhesion protein that plays an essential role in the formation of adherens junctions, and therefore in maintaining the stability and organization of epithelial lateral membranes and the maintenance of cell polarity. E-cadherin is tethered to the cytoskeleton, via its C-terminal domain, to β -catenin, forming the canonical cadherin-catenin complex to connect with the actin-based cytoskeleton. E-cadherin also directly binds to Ankyrin-G (AnkG), a scaffold protein that links membrane proteins to the spectrin-based cytoskeleton. However, the mechanism by which this complex is formed and its relationship with the cadherin-catenin complex is unknown.

Kong et al use isothermal titration calorimetry, analytical gel filtration chromatography and NMR to determine that E-cadherin binds to AnkG with a stoichiometry ranging from 1:1 to 1:2. They identify three distinct sites on E-cadherin that can bind to the same binding region on AnkG, and show that binding by the three E-cadherin sites to AnkG is dynamic in nature, with all three E-cadherin sites competing to occupy the single pocket on AnkG. The authors also show that disruption of all three bonding sites E-cadherin is necessary to abrogate binding to AnkG and furthermore, that this leads to disruption of cell polarity. Finally, the authors show that E-cadherin can bind to both AnkG and β -catenin simultaneously, providing mechanistic insights into the formation of the ankyrin-spectrin complex alongside the cadherin-catenin complex.

The study describes a previously unknown phenomenon – the promiscuous and dynamic nature of association between E-cadherin and AnkG, and the authors suggest that disrupting this phenomenon might affect cellular polarity.

However, the study so far raises several intriguing questions that have not been gone into in sufficient detail. The manuscript could be improved substantially by additional discussion/analysis/figures to flesh out few points, discussed below.

We thank the reviewer for summarizing our study and for recognition of our results. We have taken the reviewer's feedback into account and revised our manuscript accordingly. We have also included the new data and made changes in emphasis as suggested by the reviewer.

- Both E-cadherin and AnkG are associated with and function at the membrane, yet the study does not explore whether the dynamic nature of these association between these two proteins differs at a membrane. The authors show that disruption of all three binding sites on E-cadherin is necessary for disruption of cell polarity. However, this might not necessarily be (or only be) because of the disruption of the association between E-cadherin and AnkG. Interactions between the membrane and one or both proteins might also play a role. It would be good if the authors could comment on this.

We thank the reviewer for this guidance and we agree with the reviewer that micromembrane conditions play roles in regulating the interaction between membrane proteins and/or scaffold proteins. In the case of AnkG, it has been reported that palmitoylation at the membrane binding

domain allows it to insert into the plasma membranes^[9]. Moreover, AnkG interacts with numerous membrane proteins on its ANK repeats region, which is the same region where it binds to E-cadherin. In our study, we aimed to simplify the complex interactions by investigating purified protein samples to gain insights at the molecular level. Based on these *in vitro* data, we further designed our cellular experiments to demonstrate that disruption of E-cadherin-Ankyrin-G interaction resulted in the malfunction of cell polarity. The establishment and maintenance of cell polarity involve the coordination of diverse cellular pathways, and various protein complexes (*e.g.*, Par3-Par6-aPKC complex). As the reviewer mentioned, it is highly possible that the association between the membrane, E-cadherin and its binding partner (*e.g.*, AnkG) contributes to E-cadherin polarity at the lateral membrane.

- The authors have not explored the functional implications of the dynamic nature of the interaction between E-cadherin and AnkG or discussed why the dynamic nature of this interaction necessary for the maintenance of cellular polarity. It would be good to at least include a description of these implications in the Discussion section.

We appreciate the reviewer's suggestion, and we have extensively brainstormed the implications of the dynamic nature of the complex formation in the context of cellular conditions. We propose that the maintenance of the E-cadherin complex at the adhesion junctions is not solely dependent on the well-characterized cadherin-catenin complex, but rather ankyrins may play a primary role in stabilizing E-cadherin on lateral membranes. Our present study, which investigates the interface of E-cadherin and AnkG and demonstrates that their interaction is essential for the organization of lateral membranes, suggests that the dynamic nature of the interaction mechanism ensures the association between E-cadherin and AnkG under various cellular conditions. We have incorporated these points into the discussion section, as per the reviewer's suggestion.

- A few minor points:

- o In the section "Discrete sites within E-cadherin bind to AnkG in a dynamic manner", it would be good if the authors could include a description of how the various truncation fragments were determined.

Thank the reviewer for this suggestion. The determination of various truncation fragments was not performed all at once. As shown in Fig. S1, we initially divided the E-cadherin tail into two parts (734-775, 776-884) based on previous literatures reporting its interaction with p120 catenin (734-775) or beta-catenin (776-884). We found both of these two fragments could bind to AnkG. Subsequently we designed different truncations based on factors including conservation across species, amino acids properties, and the lengths of the fragments, ultimately leading to the identification of the three discrete binding sites. We have now included a description of how the truncations were determined in our revised manuscript.

- o In the section "Promiscuous engagement of E-cadherin by AnkG" the authors discuss ITC analyses of three truncations (R6-24, R8-24 and R8-14) of AnkG with the entire C-terminal domain of E-cadherin (CTD) and also with E-cadherin site 1 (residues 751-775) and E-cadherin site 2,3 (residues 828-869). However there doesn't appear to be a figure/table showing the ITC data of R8-

24 with the E-cadherin CTD. It would be good to include this.

We thank the reviewer for the careful reading and great suggestions! We have now included the ITC data of E-cadherin CTD titrated to AnkG R6-24 or AnkG R8-24 (Figure VIII). The dissociation constants (K_d) for the E-cadherin CTD with AnkG R6-24 or R8-24 are $\sim 2.9 \mu\text{M}$ and $\sim 1.5 \mu\text{M}$, respectively, which are comparable to the binding with AnkG R1-24. We have incorporated these results in our revised manuscript.

Figure VIII. ITC data show binding affinities between E-cadherin CTD and AnkG R6-24 or AnkG R8-24.

o In Figure 5a-h, it would be good to include the data from the cells expressing shRNA targeting E-cadherin, but which have not been transfected with any GFP-E-cadherin, both as a control and to verify that AnkG does not accumulate at the lateral membrane in the absence of E-cadherin.

We thank the reviewer for this helpful guidance. We present these data in Figure S8c in our original manuscript (Figure IXa). Inspired by the reviewer, we re-quantified the AnkG accumulation at the lateral membrane (Figure IXb) and the lateral membrane height (Figure IXc) in E-cadherin knockdown cells transfected with GFP vector as the control and updated our quantification figures in the revised manuscript. We have now revised our manuscript accordingly.

Figure IX. (a) MDCK cells stably expressing a shRNA targeting E-cadherin were transfected with the GFP vector as a control and stained with AnkG and actin signals (related to Figure 5). Scale bars: 10 μm . XZ projection is shown at bottom. Scale bars: 5 μm . (b-c) Quantification of the immunofluorescence intensity ratios of cytosolic to membranes for AnkG in the groups of WT (n=41); site-1 to A (n=40); site-2 to A (n=37); site-3 to A (n=44); site-1, 2 to A (n=43); site-1, 3 to A (n=53); site-2, 3 to A (n=55); site-1, 2, 3 to A (n=43); and GFP control (n=25) (b), and the lateral membrane height in the groups of WT (n=62); site-1 to A (n=68); site-2 to A (n=56); site-3 to A (n=57); site-1, 2 to A (n=61); site-1, 3 to A (n=58); site-2, 3 to A (n=61); site-1, 2, 3 to A (n=65); and GFP control (n=43) (c). Data are presented as means \pm SEM and analyzed using one way ANOVA followed by Dunnett's multiple comparisons test to WT. * $p < 0.05$, **** $p < 0.0001$, ns, not significant.

References

- [1] Jenkins P M, Vasavda C, Hostettler J, et al. E-cadherin polarity is determined by a multifunction motif mediating lateral membrane retention through ankyrin-G and apical-lateral transcytosis through clathrin [J]. *J Biol Chem*, 2013, 288(20): 14018-31.
- [2] Kim J H, Liao D, Lau L F, et al. SynGAP: a synaptic RasGAP that associates with the PSD-95/SAP90 protein family [J]. *Neuron*, 1998, 20(4): 683-91.
- [3] Cheng D, Hoogenraad C C, Rush J, et al. Relative and absolute quantification of postsynaptic density proteome isolated from rat forebrain and cerebellum [J]. *Mol Cell Proteomics*, 2006, 5(6): 1158-70.
- [4] Capaldo C T, Macara I G. Depletion of E-cadherin disrupts establishment but not maintenance of cell junctions in Madin-Darby canine kidney epithelial cells [J]. *Mol Biol Cell*, 2007, 18(1): 189-200.
- [5] Kizhatil K, Davis J Q, Davis L, et al. Ankyrin-G is a molecular partner of E-cadherin in epithelial cells and early embryos [J]. *J Biol Chem*, 2007, 282(36): 26552-61.
- [6] Stabach P R, Devarajan P, Stankewich M C, et al. Ankyrin facilitates intracellular trafficking of alpha1-Na⁺-K⁺-ATPase in polarized cells [J]. *Am J Physiol Cell Physiol*, 2008, 295(5): C1202-14.
- [7] Wang C, Wei Z, Chen K, et al. Structural basis of diverse membrane target recognitions by ankyrins [J]. *Elife*, 2014, 3.
- [8] He L, Jiang W, Li J, et al. Crystal structure of Ankyrin-G in complex with a fragment of Neurofascin reveals binding mechanisms required for integrity of the axon initial segment [J]. *J Biol Chem*, 2022, 298(9): 102272.
- [9] He M, Jenkins P, Bennett V. Cysteine 70 of ankyrin-G is S-palmitoylated and is required for function of ankyrin-G in membrane domain assembly [J]. *J Biol Chem*, 2012, 287(52):43995-4005.

REVIEWER COMMENTS

Reviewer #1 (Remarks to the Author):

The new manuscript has addressed my concerns, and I believe that this important paper is ready for publication.

Reviewer #2 (Remarks to the Author):

The revised manuscript contains little new data and only minor changes to the text. The authors addressed most of my points in the rebuttal letter, but did not follow them with substantive changes to the manuscript. Thus, I still have concerns about the breadth and impact of the study.

I remain particularly concerned about the significance of the E-cadherin/Ankyrin interaction in vivo. The authors offered a rebuttal but little/no new data to bolster their claims. I am also still puzzled by the imaging results in the Ecad knockdown/rescue experiments in Figure 5. This figure is especially important as it is the only data offering cell biological relevance to the biochemical data. For example, they show that b-catenin and p120-catenin levels are unaffected in the stable E-cadherin knockdown line and compare it to work from Makara lab (who also observed no change in catenin recruitment upon E-cadherin knockdown) However, the Makara paper focused on junction maintenance and examined catenin localization in cells that were allowed to form junctions and THEN depleted of E-cadherin (transient knockdown). These are two different experimental approaches and the comparison is not appropriate. The p120/b-cat/a-cat localization data in the Ecad knockdown and full-length E-cad rescued cells should be quantified and included in Fig 5. If significant differences are observed, catenin recruitment should be analyzed for all mutants.

Reviewer #3 (Remarks to the Author):

Authors have responded to the questions/comments raised by the reviewers. The dynamic nature of binding of ankyrin-G with the cytoplasmic domain of E-cadherin is elucidated using bioanalytical and structural biology methods. The manuscript can be accepted for publication in Nature Communications.

Point-by-point responses to the comments raised by the reviewers

Before point-by-point responses to the referees' comments, we thank all the reviewers for recognizing the novelty and interests of our works and their critical and constructive suggestions and guidance that help us to efficiently improve our manuscript. Our responses are shown in blue.

Reviewer #1 (Remarks to the Author):

The new manuscript has addressed my concerns, and I believe that this important paper is ready for publication.

We thank the reviewer for the support.

Reviewer #2 (Remarks to the Author):

The revised manuscript contains little new data and only minor changes to the text. The authors addressed most of my points in the rebuttal letter, but did not follow them with substantive changes to the manuscript. Thus, I still have concerns about the breadth and impact of the study.

I remain particularly concerned about the significance of the E-cadherin/Ankyrin interaction in vivo. The authors offered a rebuttal but little/no new data to bolster their claims. I am also still puzzled by the imaging results in the Ecad knockdown/rescue experiments in Figure 5. This figure is especially important as it is the only data offering cell biological relevance to the biochemical data. For example, they show that b-catenin and p120-catenin levels are unaffected in the stable E-cadherin knockdown line and compare it to work from Makara lab (who also observed no change in catenin recruitment upon E-cadherin knockdown) However, the Makara paper focused on junction maintenance and examined catenin localization in cells that were allowed to form junctions and THEN depleted of E-cadherin (transient knockdown). These are two different experimental approaches and the comparison is not appropriate. The p120/b-cat/a-cat localization data in the Ecad knockdown and full-length E-cad rescued cells should be quantified and included in Fig 5. If significant differences are observed, catenin recruitment should be analyzed for all mutants.

We thank the reviewer for these helpful suggestions and guidance on comparing the previous literature with our current data. We totally agreed with the reviewer that examinations of the p120/b-cat/a-cat localizations in the E-cad knockdown and rescued cells will provide further biological relevance. Following the reviewer's guidance, we conducted new experiments using the same stable cell lines and protocols. In summary, we stained the endogenous β -catenin, α -catenin and p120-catenin signals in the stable cell lines rescued with WT E-cadherin, control vector (GFP control), and our E-cadherin variants (Fig. I). Interestingly, the results showed that there were no obvious effects on β -catenin, α -catenin, or p120 catenin localizations and they remained the membrane accumulation in cells depleting E-cadherin (Fig. I), showing distinct behaviors compared with AnkG's localization. One of the potential mechanisms may be that MDCK cells express several other classical cadherins, e.g. cadherin-6 (K-cadherin), a type II

atypical cadherin. When we knocked down E-cadherin, these catenin proteins are stabilized by these classical cadherins, especially the K-cadherin. In contrast, AnkG mislocalized in cytosol when depleting E-cadherin, presenting the different binding mechanisms described *in vitro* may contribute the different cellular phenomenon. We have incorporated these data into our revised manuscript. We believe these new data will provide more information regarding the lateral membrane organizations mediated by E-cadherin complex.

Fig. 1 Catenin proteins preserve membrane localizations when E-cadherin is knocked down in MDCK cells. (a-i) MDCK cells stably expressing a shRNA targeting were transfected with wild-type (WT) GFP-E-cadherin (a), GFP-E-cadherin site-1 to A (b), GFP-E-cadherin site-2 to A (c), GFP-E-cadherin site-3 to A (d), GFP-E-cadherin site-1, 2 to A (e), GFP-E-cadherin site-1, 3 to A (f), GFP-E-cadherin site-2, 3 to A (g), GFP-E-cadherin site-1, 2, 3 to A (h), and GFP vector (i) respectively. The cells were stained with GFP signals showing E-cadherin localization, endogenous β -catenin and α -catenin signals showing β -catenin and α -catenin localization (Scale bars: 10 μ m.). (j-r) MDCK cells stably expressing a shRNA targeting were transfected with WT GFP-E-cadherin (j), GFP-E-cadherin variants (k-q), and GFP vector (r) respectively. Endogenous p120 catenin signal showing p120 catenin localization (Scale bars: 10 μ m.). (s) Quantification of the immunofluorescence intensity ratios of cytosolic β -catenin vs plasma membranes for E-cadherin WT (n=46); site 1 to A (n=43); site 2 to A (n=50); site 3 to A (n=35); site 1, 2 to A (n=37); site 1, 3 to A (n=40); site 2, 3 to A (n=44); site 1, 2, 3 to A (n=41); GFP vector (n=36). (t) Quantification of the immunofluorescence intensity ratios of cytosolic α -catenin vs plasma membranes for E-cadherin WT (n=44); site 1 to A (n=43); site 2 to A (n=44); site 3 to A (n=46); site 1, 2 to A (n=50); site 1, 3

to A (n=50); site 2, 3 to A (n=48); site 1, 2, 3 to A (n=46); GFP vector (n=47). (u) Quantification of the immunofluorescence intensity ratios of cytosolic p120 catenin vs plasma membranes for E-cadherin WT (n=47); site 1 to A (n=47); site 2 to A (n=42); site 3 to A (n=42); site 1, 2 to A (n=43); site 1, 3 to A (n=38); site 2, 3 to A (n=47); site 1, 2, 3 to A (n=43); GFP vector (n=44). Data are presented as means \pm SEM and analyzed using one-way ANOVA followed by Dunnett's multiple comparisons test to β -catenin (s), α -catenin (t), and p120 catenin (u). ns, not significant.

Reviewer #3 (Remarks to the Author):

Authors have responded to the questions/comments raised by the reviewers. The dynamic nature of binding of ankyrin-G with the cytoplasmic domain of E-cadherin is elucidated using bioanalytical and structural biology methods. The manuscript can be accepted for publication in Nature Communications.

We thank the reviewer for the support.

REVIEWERS' COMMENTS

Reviewer #2 (Remarks to the Author):

I appreciate the authors addressing my concerns about the cell biology experiments. I am satisfied with the new quantification.

A couple of final notes:

In Fig 6, I assume the striking difference in cell area in the GFP control and some of the Ecad mutants is due to the lack of polarization. It might be worth mentioning this in the Results.

I caught at least one figure reference in the text that lacked a figure number. The authors should double check all figure references.

Point-by-point responses to the comments raised by the reviewers

Before point-by-point responses to the referees' comments, we thank all the reviewers for recognizing the novelty and interests of our works and their critical and constructive suggestions and guidance that help us to efficiently improve our manuscript. Our responses are shown in blue.

Reviewer #2 (Remarks to the Author):

I appreciate the authors addressing my concerns about the cell biology experiments. I am satisfied with the new quantification.

A couple of final notes:

In Fig 6, I assume the striking difference in cell area in the GFP control and some of the Ecad mutants is due to the lack of polarization. It might be worth mentioning this in the Results.

We totally agree with the reviewer. We have now revised our manuscript by noting this point in the results part.

I caught at least one figure reference in the text that lacked a figure number. The authors should double check all figure references.

We thank the reviewer for pointing out our oversight. We have now corrected all figure references in our manuscript and checked our manuscript throughoutly.